# The ZIKV NS5 Protein Aberrantly Alters the Tubulin Cytoskeleton, Induces the Accumulation of Autophagic p62 and Affects IFN Production: HDAC6 Has Emerged as an Anti-NS5/ZIKV Factor

**DOI:** 10.3390/cells13070598

**Published:** 2024-03-29

**Authors:** Silvia Pérez-Yanes, Iria Lorenzo-Sánchez, Romina Cabrera-Rodríguez, Jonay García-Luis, Rodrigo Trujillo-González, Judith Estévez-Herrera, Agustín Valenzuela-Fernández

**Affiliations:** 1Laboratorio de Inmunología Celular y Viral, Unidad de Farmacología, Sección de Medicina, Facultad de Ciencias de la Salud, Universidad de La Laguna, 38200 La Laguna, Spain; sperezya@ull.edu.es (S.P.-Y.); ilorenzs@ull.edu.es (I.L.-S.); rcabrerr@ull.edu.es (R.C.-R.); jgarcial@ull.edu.es (J.G.-L.); 2Department of Análisis Matemático, Facultad de Ciencias, Universidad de La Laguna, 38296 La Laguna, Spain; rotrujil@ull.edu.es

**Keywords:** ZIKV, NS5, aberrant MTs, MT acetylation, autophagic p62, HDAC6, tubulin deacetylase, BUZ domain, NS5 clearance, inhibition of IFN production

## Abstract

Zika virus (ZIKV) infection and pathogenesis are linked to the disruption of neurogenesis, congenital Zika syndrome and microcephaly by affecting neural progenitor cells. Nonstructural protein 5 (NS5) is the largest product encoded by ZIKV-RNA and is important for replication and immune evasion. Here, we studied the potential effects of NS5 on microtubules (MTs) and autophagy flux, together with the interplay of NS5 with histone deacetylase 6 (HDAC6). Fluorescence microscopy, biochemical cell-fractionation combined with the use of HDAC6 mutants, chemical inhibitors and RNA interference indicated that NS5 accumulates in nuclear structures and strongly promotes the acetylation of MTs that aberrantly reorganize in nested structures. Similarly, NS5 accumulates the p62 protein, an autophagic-flux marker. Therefore, NS5 alters events that are under the control of the autophagic tubulin-deacetylase HDAC6. HDAC6 appears to degrade NS5 by autophagy in a deacetylase- and BUZ domain-dependent manner and to control the cytoplasmic expression of NS5. Moreover, NS5 inhibits RNA-mediated RIG-I interferon (IFN) production, resulting in greater activity when autophagy is inhibited (i.e., effect correlated with NS5 stability). Therefore, it is conceivable that NS5 contributes to cell toxicity and pathogenesis, evading the IFN-immune response by overcoming HDAC6 functions. HDAC6 has emerged as an anti-ZIKV factor by targeting NS5.

## 1. Introduction

The *Flaviviridae* (genus *Flavivirus*) Zika virus (ZIKV) is a mosquito-borne, positive-sense single-stranded RNA (RNA+) virus [1,2,3,4,5,6,7,8,9,10,11,12]. ZIKV is classified by homology to the Spondweni virus (SPONV) in the Spondweni viral clade or serogroup [2,13,14]. Notably, ZIKV and SPONV viruses were first characterized in Africa in 1947 and 1952 [11,15], respectively. ZIKV was discovered during the search for a potential vector responsible for the cycle of sylvan yellow fever virus (YFV) in Uganda [16] and was found in the serum of a pyrexial rhesus monkey caged in the canopy of the Zika Forest in Uganda [11].

The first confirmed human infections by ZIKV occurred in Nigeria (1954) [17], followed by several cases reported in Uganda (1962–1963) [18] and in central Java, Indonesia (1977) [19]. The World Health Organization (WHO) declared Zika an “extraordinary event that needed a coordinated response, constituting a public health emergency of international concern (PHEIC)” [20], due to the description of a large outbreak of rash in patients who were ill [21,22,23] and had short-term and low-grade fever [23] (however, this was not in all cases), and a cluster of microcephaly in newborns of infected mothers [24,25,26,27,28,29,30,31,32], together with neurological abnormalities and Guillain-Barré syndrome (GBS) in Brazil [21,33,34,35,36,37,38,39,40].

In fact, ZIKV infection carries the risk of adverse pregnancy outcomes, including increased risk of preterm birth, foetal death and stillbirth, and congenital malformations collectively characterized as congenital Zika syndrome (CZS). These include the abovementioned microcephaly, abnormal brain development, limb contractures, eye abnormalities, brain calcifications, and other neurologic manifestations [27,41,42,43,44,45]. ZIKV has been found in the cerebrospinal fluid (CSF) and brain of adults infected by the virus, who manifested neurological disorders [34,46,47,48,49]. This flavivirus causes harmful effects in the adult brain, such as GBS [46,48,50,51,52], encephalitis [46,51,52,53], meningoencephalitis [34,54], acute myelitis [46,52,55] and encephalomyelitis [46,49,51,56,57], as well as sensory polyneuropathy [58] and other neurological complications [59,60].

Several ZIKV proteins have been well characterized for their functions in the pathology of the virus, as well as for their biology (i.e., viral infection and transmission) and immune escape [61,62,63,64,65,66,67,68,69,70,71,72,73,74,75,76,77,78,79,80,81,82,83,84,85,86,87]. Furthermore, microtubules (MTs) play a critical role in ZIKV infection, as in other flavivirus life cycles, and are associated with the modification of the tubulin cytoskeleton and the closely related process of autophagy to the pathogenesis of the virus [88,89,90,91,92,93,94,95]. In this sense, nonstructural protein 5 (NS5) of ZIKV appears to interact with components of the cilium base that are associated with MTs, promoting ciliopathy and premature neurogenesis [96]. However, little is known about the functional involvement of the NS5 protein in altering tubulin cytoskeleton dynamics or sequestosome 1 (SQSTM1)/p62-associated autophagy flux [97,98,99,100,101,102,103]; these are key events for cell permissivity and survival that are regulated by the antiviral factor histone deacetylase 6 (HDAC6) [91,94,98,99,100,101,104,105,106,107,108].

Considering the accepted role of MTs in innate immunity and infection [92,109,110,111,112], their essential roles in intracellular trafficking and cell morphology (reviewed in [113]), the involvement of the ZIKV NS5 protein in virus replication and evasion from the interferon (IFN)-associated immune response, and its association with the centrosome or microtubule (MT) organized centre (MTOC) during mitosis and cell division [95,114,115,116], we aimed to study the effect of the ZIKV NS5 protein on MTs to ascertain potential interplay at this level. Our results indicate that the ZIKV NS5 protein mainly accumulates in nuclear structures, as previously reported [64,65,117,118], and promotes the acetylation of MTs that aberrantly reorganize in nested structures organized at the cell periphery. Similarly, we observed that the p62 protein—a marker of autophagy flux [97,98,99,100,101,102,103]—accumulated in cells overexpressing the ZIKV NS5 protein. Notably, these data indicate that NS5 alters cell events that are under the control of the antiviral tubulin deacetylase and the autophagy-associated enzyme HDAC6 [91,98,99,100,101,104,105,106,107,108]. HDAC6 appears to degrade NS5 by autophagy in a deacetylase- and BUZ (binder of or bound to ubiquitin zinc finger [119,120]) domain-dependent manner and by controlling the cytoplasmic expression of NS5. Similarly, our results indicate that the ZIKV NS5 protein inhibits foreign RNA-mediated retinoic acid-inducible gene I (RIG-I) interferon (IFN) production, resulting in increased activity when autophagy is inhibited.

Taken together, these results suggest that the ZIKV NS5 protein contributes to cell toxicity and pathogenesis by affecting MT dynamics and p62-associated autophagic flux and evades the IFN-immune response by overcoming HDAC6 functions. Therefore, HDAC6 has emerged as an anti-ZIKV factor by targeting the ZIKV NS5 protein.

## 2. Materials and Methods

### 2.1. Antibodies and Reagents

Mouse anti-myc (9E10; sc-40), rabbit anti-HDAC6 (H-300; sc-11420), rabbit anti-HA (Y-11; sc-805), mouse anti-p62/SQSTM1 (D3; sc-28359) and mouse anti-Histone 3 (1G1; sc-517576) were obtained from Santa Cruz Biotechnology (Santa Cruz, CA, USA). Rabbit anti-NS5 (GTX133312) was obtained from GeneTex (GeneTex, Irvine, CA, USA). Mab anti-α-tubulin (T6074), anti-acetylated α-tubulin (T7451), and secondary horseradish peroxidase (HRP)-conjugated Abs specific for any Ab species assays were purchased from Sigma-Aldrich (Sigma-Aldrich, St. Louis, MO, USA). Acetyl-α-tubulin (Lys40) (D20G3) XP^®^ rabbit mAb (5335S) was acquired from Cell Signaling Technology (Cell Signaling Technology, Danvers, MA, USA). Alexa Fluor 568-conjugated goat α-mouse IgG1 (γ1) (A21124) and Alexa Fluor 488-conjugated goat α-mouse IgG1 (γ1) (A21121) were obtained from Life Technologies (Life Technologies Corporation, CA, USA). FuGENE HD Transfection Reagent (E2312) and Lipofectamine^TM^ RNAi MAX (13778075) were purchased from Promega (Promega Corporation, Fitchburg, WI, USA) and Thermo Fisher (Thermo Fisher Scientific, Waltham, MA, USA), respectively. Mission^®^ siRNA Universal Negative Control #1 (SIC001), 3-methyladenine (3-MA) (M9281), tubacin (SML0065), 4′,6-diamidino-2-phenylindole (DAPI), phenylmethylsulfonyl fluoride (PMSF) and poly-D-lysine (P6407) were obtained from Sigma-Aldrich (Sigma-Aldrich). A Complete™ protease inhibitor cocktail (11697498001) was obtained from Roche Diagnostics (GmbH, Mannheim, Germany).

### 2.2. DNA Plasmids and Viral DNA Constructs

The DNA sequence encoding nonstructural protein 5 (NS5) of ZIKV (Zika strain MP1751_East_African (Uganda)) was cloned with the myc or human influenza hemagglutinin (HA) epitope at the C-terminus using the BamHI/XhoI restriction sites and T4 DNA ligation in pcDNA^TM^ 3.1(+). HDAC6 constructs were generated by Drs. X.-J. Yang and N. R. Bertos (Molecular Oncology Group, Department of Medicine, McGill University Health Centre, Montreal, QC, Canada) [101,106,121], and the HA epitope was introduced as an N-terminal tag (HA-wt-HDAC6), as previously reported [100,101,106]. When indicated, we used double mutations affecting the HDAC6 catalytic domain (H216A/DD1 mutant and H611A/DD2 mutant (HA-dm-HDAC6)) or deletions in the proautophagic BUZ domain (HA-HDAC6-∆BUZ), which lacks the 1092 residue in the C-terminal region that bears the Cys/His-rich motif [101,106,120,121,122]. These constructs are unable to promote the autophagic degradation of some viral proteins, as previously reported [100,101,106]. The wild-type (wt)-HDAC6 encoding the fluorescent protein DsRed was obtained using the AgeI/NotI restriction sites of pDsRed (Clontech, Palo Alto, CA, USA), as previously reported [100]. The pcDNA^TM^ 3.1(+) empty vector (Life Technologies) was used as a control for cDNA transfection.

### 2.3. Cells

HEK-293T cells (cat. number 103, NIH AIDS Research and Reference Reagent Program) and Lucia luciferase reporter HEK-293 cells expressing human RIG-I, HEK-Lucia™ RIG-I, (hkl-hrigi, InvivoGen, San Diego, CA, USA) were grown at 37 °C in a humidified atmosphere with 5% CO_2_ in DMEM (Lonza, Verviers, Belgium) supplemented with 10% foetal calf serum (FCS) (Lonza), 1% L-glutamine and 1% penicillin–streptomycin (Lonza, Basel, Switzerland). The cells were harvested and passaged every 3 days using trypsin-EDTA (L0930-100) (Biowest, Lakewood Ranch, FL, USA) or Versene 1× (15040-066; Gibco Chemicals, Thermo Fisher Scientific, MA, USA). HEK-Lucia™ RIG-I cells were cultured to 50–70% confluence in fresh supplemented DMEM 24 h before cell transfection with viral or human DNA constructs and before the induction of interferon (IFN) production. To maintain the stable expression of RIG-I and the luciferase reporter downstream of tandem interferon-stimulated gene 54 (ISG54) promoter elements, 30 μg/mL blasticidin and 100 μg/mL Zeocin™ were added to the HEK-Lucia™ RIG-I reporter cell line. Mycoplasma-free (Mycozap antibiotics, Lonza) or 100 μg/mL Normocin™ was routinely added to each HEK-293T and HEK-Lucia™ RIG-I cell split. Cell viability was quantitatively determined by light microscopic quantitation visualizing trypan blue-stained cells under each experimental condition.

### 2.4. Messenger RNA Silencing

We used short interfering RNA (siRNA) oligonucleotides (oligos) specifically generated by Eurogentec (Southampton, UK) against the mRNA sequence of HDAC6 to knockdown HDAC6 expression. Transient siRNA transfections were performed using Lipofectamine^TM^ RNAi MAX dissolved in Opti-MEM^®^. A mixture of Lipofectamine^TM^ RNAi MAX/oligos (1:1) was gently vortexed, incubated for 5 min at room temperature (RT), and then added to HEK-293T cells cultured in 6-well plates. A total of 25 pmol/well of each commercial scrambled control oligo (Mission^®^ siRNA Universal Negative Control #1) (Sigma-Aldrich) or HDAC6-specific siRNA oligos spanning nucleotides 193–213, 217–237 and 284–304 were used as previously described [100,101,106]. After 24 h, the cells were transfected with siRNAs, and the same cells were subjected to a second round of treatment with FuGENE HD Transfection reagent for transient expression of viral NS5 or pcDNA^TM^ 3.1(+) empty vector under control conditions and incubated for 24 h to complete a 48 h interval of HDAC6 silencing, when the siRNAs for HDAC6 induced specific interference with protein expression. Cells were then lysed and analysed with specific Abs by Western blotting to validate endogenous HDAC6 silencing, as well as the associated expression of the studied cell and viral proteins, as similarly described [98,99,100,101,106].

### 2.5. Cell Fractioning

HEK-293T cells (1 × 10^6^) were harvested 48 h post transfection by using a cell scraper, washed with PBS and pelleted at 200× *g* (1500 rpm) for 5 min. The cells were resuspended by gentle pipetting in cytosolic extraction buffer (CEB) (10 mM HEPES pH 7.5, 10 mM KCl, 0.1 mM EDTA, 1 mM dithiothreitol (DTT), 0.5% Nonidet-40 and 0.5 mM PMSF), usually approximately 80–100 μL (5 times the volume of pellet), and incubated for 20 min on ice (+4 °C), pipetting after 10 min to promote the cell lysis process. Centrifugation was performed at 12,000× *g* (11,400 rpm) at +4 °C for 10 min, and the resulting supernatant containing the cytoplasmic extract was separated into another tube (Eppendorf-like size) and washed three times (centrifugation between washes was performed for 5 min at +4 °C at 12,000× *g*). The pellet corresponding to the nuclear extract was washed at least twice with CEB to eliminate cytoplasmic debris by centrifugation at 12,000× *g* at +4 °C for 5 min and resuspended in 50 μL (2 times the volume of pellet, taking into account the nucleus-cytoplasm ratio) of nuclear extraction buffer (NEB) (20 mM HEPES pH 7.5, 400 mM NaCl, 1 mM EDTA, 1 mM DTT, 1 mM PMSF) and incubated on ice for 30 min; vortexing took place once every 10 min. Finally, the pellet was centrifuged for 15 min at +4 °C and 12,000× *g*, and the supernatant was collected as the nuclear fraction. For Western blot analysis, α-tubulin was used as the cytoplasmic marker and for control of total protein load, and histone 3 was used as the marker for the nuclear fraction.

### 2.6. Western Blotting

Protein expression in cell lysates was determined by SDS–PAGE and Western blotting. HEK-293T or HEK-Lucia™ RIG-I cells were transfected with different cDNA constructs using FuGENE^®^ HD Transfection reagent in a mixture of FuGene/plasmids at a 3:1 ratio, gently vortexed, incubated for 10 min at RT, and then added to the cells in culture. Briefly, 48 h after transfection, the cells were treated with lysis buffer (50 mM Tris-HCl (pH 7.5), 150 mM NaCl, 1% Triton X-100, 0.5% sodium deoxycholate and 1× Complete™ Protease Inhibitor Cocktail) for 30 min and sonicated for 30 s at +4 °C. The effects of the different inhibitors were similarly assayed in HEK-293T cells or HEK-Lucia™ RIG-I cells, as indicated in each case. To inhibit HDAC6 catalytic activity, cells were treated overnight at 37 °C with 1 μM tubacin in DMSO (<0.1% of the total volume). To prevent autophagosome formation and subsequent autophagic degradation, 3-MA (5 mM) in PBS was added for 5 h before cell lysis. Equivalent amounts of protein (30–40 μg), which were determined using the bicinchoninic acid (BCA) method (Millipore Corporation, Billerica, MA, USA), were resuspended and treated with Laemmli buffer and then separated by 10–12% SDS–PAGE and electroblotted onto 0.45 μm-polyvinylidene difluoride (PVDF) membranes (Millipore) using Trans-blot Turbo (Bio-Rad, Hercules, CA, USA). The membranes were blocked with 5% non-fat dry milk in TBST (100 mM Tris, 0.9% NaCl, pH 7.5, 0.1% Tween 200) for 30 min and then incubated with specific Abs. Proteins were detected by luminescence using an enhanced chemiluminescence (ECL) system (Bio-Rad) and analysed using a ChemiDoc MP device and Image LabTM Software, version 5.2 (Bio-Rad).

### 2.7. Fluorescence Microscopy

HEK-293T cells (1 × 10^5^ cells) were seeded on poly-D-lysine-coated Ø 18 mm glass coverslips and transfected with NS5-myc or cotransfected with the different pcDNA^TM^ 3.1(+) empty vector (control) or wt-HDAC6-DsRed plasmids using FuGENE^®^ HD Transfection Reagent as described above to analyse NS5 expression and distribution. After 48 h post transfection, the cells were washed three times with PBS, fixed for 15 min in 4% paraformaldehyde (in PBS), permeabilized for 45 min with PBS-T (0.1% Tween 20 in 1× PBS), quenched with 100 mM glycine in 1× PBS for 15 min and treated with blocking solution (5% FCS in PBS-T) for up to 1 h. Mouse α-myc (9E10; sc-40) (1:50) and Alexa Fluor 568 goat α-mouse IgG1 (γ1) or Alexa Fluor 488 goat α-mouse IgG1 (γ1) (1:500) were subsequently incubated overnight or for 1 h, after which they were diluted in blocking solution. Coverslips were then washed and mounted in Mowiol Antifade (Dako, Glostrup, Denmark), and image acquisition was performed by epifluorescence microscopy (Zeiss 200 M (Zeiss, Jena, Germany)), as previously reported [100,123,124]. The expression and distribution of fluorescent NS5 were analysed via a line scan and quantified using MetaMorph software (http://visualdynamix.net/pdf/imagingsoftware/MetaMorph.pdf (accessed on 22 March 2024); Universal Imaging, Downington, PA, USA), as previously described [100,123,124].

### 2.8. RIG-I/ISG54 Activity Assay: RNA-Mediated RIG-I IFN Production

HEK-Lucia™ RIG-I cells were transfected with the NS5 construct or control plasmid, as described above for HEK-293T cells. Depending on the experimental conditions, at 48 h post transfection, some cells were treated with 200 mM 3-MA, an autophagy inhibitor, for 5 h, and PBS (a vehicle for 3-MA) was used as a control. HEK-Lucia™ RIG-I cells were then detached using Versene (Gibco, Waltham, MA, USA), and a fraction of the cells was used for lysis, while the remaining cells were seeded in a 96-well plate at a concentration of 1 × 10^5^ cells/well in 180 μL of antibiotic-free media (DMEM supplemented with 10% FBS) (six replicates/condition). The RIG-I agonist, 5′ triphosphate hairpin RNA (3p-hpRNA) and RIG-I-like receptor ligands (tlrl-hprna, InvivoGen) were transfected with LyoVec™ (Lyec-2, InvivoGen) following the manufacturer’s instructions, and the cells were incubated overnight to ensure adequate induction. The next day, 20 μL of the supernatant was tested for ISG54-Lucia secretion, whose expression depended on the activation of the IFN-I promoters, using 50 μL of QUANTI-Luc™ Gold (rep-qlcg-2), which contains coelenterazine and allows the Lucia signal to be measured, added by automatic injection into the wells of a 96-well white microplate. Luminescence was measured by obtaining relative light units (RLUs) for each condition at the same time point after injection using a Luminoskan^TM^ Ascent microplate luminometer (Thermo Fisher Scientific, MA, USA).

### 2.9. Statistical Analysis

Statistical analyses were performed using GraphPad Prism, version 6.0b (GraphPad Software, San Diego, CA, USA). The significance of differences between groups was determined using Student’s *t* test, as indicated in the figure legends.

## 3. Results

Considering the above findings, we investigated the effect of the ZIKV NS5 protein on cell MTs, the interplay of ZIKV NS5 with potential associated factors, such as the tubulin deacetylase HDAC6, and specific RNA/RIG-I-triggered IFN production.

### 3.1. The ZIKV NS5 Protein Mainly Localizes to Nuclear Structures

First, we examined the subcellular localization of the ZIKV NS5 protein in HEK-293T cells, a model for ZIKV and other RNA virus infection and production [98,99,100,101,125,126,127], which were transfected with the NS5-myc construct. At 48 h post transfection (Figure 1a), ZIKV NS5 protein was mainly localized to the nucleus, as observed by biochemical analysis of cell fractioning (Figure 1b) and by immunofluorescence of transfected cells (Figure 1c).

Therefore, ZIKV NS5 appears to be restricted to nuclear structures, as previously reported [64,65,117,118], according to fluorescence microscopy analysis (Figure 1c). This observation was confirmed by cell-fraction biochemical analysis of cells overexpressing the ZIKV NS5 protein, which is mainly distributed in the nuclear fractions and is weakly detected in the cytoplasmic fractions (Figure 1b). Hence, we observed that ZIKV NS5 localized to discrete punctate nuclear bodies. NS5 appears as regular 20–50 punctate spot-shaped structures that vary in size from one (Figure 1c, type i) to several (Figure 1c, type ii) micrometres in diameter. Notably, mostly irregularly shaped structures that range in length form ring-like structures that we define as “speckles” (Figure 1c, type iii). The diameter observed for the different NS5 structures (i.e., line scan analysis), their frequency per cell (%) and the amount of NS5 detected are shown (Figure 1d, Figure 1e and Figure 1f, respectively). This punctate localization observed by immunofluorescence microscopy presented appreciably less intense DAPI staining in the structure core, suggesting the absence of any DNA colocalization, as similarly reported [65]. Although the cells were synchronously transfected, the ZIKV NS5 protein selectively formed diverse structural patterns in different cells, showing its highly dynamic nature and governed by processes that are independent of time, as previously reported in studies carried out with infectious viruses in which diversity was attributed to distinct phases of the viral cycle [64].

Under these experimental conditions, our results indicate that the NS5 protein mainly accumulated in the nucleus and behaved in a very dynamic manner, residing in entities whose number and size varied, suggesting nuclear accumulation of the ZIKV NS5 protein.

### 3.2. The ZIKV NS5 Protein Promotes the Acetylation of Microtubules That Reorganize into a Nested Structure at the Cell Periphery

MTs have been reported to support ZIKV and other flavivirus infections [88,95,128]. In fact, the ZIKV NS5 protein was reported to be localized to the MTOC or centrosome during mitosis and cell division [95,114,115,116]. Hence, we aimed to study the effect of the ZIKV NS5 protein on MTs to ascertain the potential interplay at this level.

Fluorescence microscopy studies of HEK-293T cells overexpressing the NS5-myc construct indicated that the presence of the ZIKV protein promoted the aberrant reorganization of MTs, which form a caged or nested structure at the periphery of cells (Figure 2a,b). The MTs of this caged structure are highly acetylated at the Lys^40^ residue of the α-tubulin subunits of MTs (Figure 2a,b), a post-translational modification associated with stable MTs [105,106,129,130,131,132,133]. HDAC6 is an antiviral cytoplasmic enzyme [91,98,100,101,104,105,106] that regulates MT dynamics by deacetylating the Lys^40^ residue of the α-tubulin subunit of stabilized MTs [107,108,134]. We therefore compared the effect of a specific HDAC6 deacetylase inhibitor, tubacin [107,108,134,135,136], with that of NS5 on the acetylation of MTs (Figure 2b). The results obtained indicate that tubacin-mediated inhibition of the HDAC6 tubulin-deacetylase promotes acetylation of MTs, as similarly observed with the ZIKV NS5 protein. Notably, tubacin does not induce the aberrant reorganization of MTs in nested peripheric structures that occurs in cells expressing NS5 (Figure 2b). Biochemical analysis of these NS5-positive cells revealed that NS5 promoted the acetylation of MTs in a dose-dependent manner (Figure 2c, quantified in the associated histograms). These data indicate that NS5 stably and aberrantly reorganizes cellular MTs, provoking their acetylation and overcoming the potential protective effect of the endogenous tubulin deacetylase HDAC6.

It has been reported that HDAC6 exerts its antiviral activity against several key human immunodeficiency virus type 1 (HIV-1) factors (i.e., the envelope complex (Env), Vif and Pr55^Gag^ proteins [98,99,100,101,105,106]) through its tubulin-deacetylase activity and the C-terminal BUZ domain (also known as polyubiquitin associated zinc finger (PAZ)) [119] or ubiquitin carboxyl-terminal hydrolase-like zinc finger, ubiquitin-specific protease (ZnF-UBP) domain [137]), both of which are necessary for autophagy-mediated viral protein clearance [100,101]. We investigated whether the ZIKV NS5 protein could disturb autophagy, as measured by the level of expression of the p62 protein, a marker of autophagic p62 flux that interacts and works with HDAC6 in the autophagic clearance of proteins, where p62 also fades [97,98,99,100,101,102,103]. The ZIKV NS5 protein negatively altered autophagic p62 flux since cells expressing NS5 accumulated p62 protein (Figure 2d), indicating that NS5 had a dose-dependent effect on the accumulation of the autophagy marker p62 (Figure 2e).

### 3.3. HDAC6 Targets the ZIKV NS5 Protein via Autophagy and Requires Its Deacetylase Activity and BUZ Domain for NS5 Clearance

Considering the above data, the ZIKV NS5 protein appears to alter HDAC6-associated substrates that are key for the antiviral functions of the enzyme, such as MT dynamics (i.e., organization and acetylation) and the stability of the autophagic p62 factor [91,98,99,100,101,105,106].

We next studied the effect of HDAC6 on the ZIKV NS5 protein. Biochemical (Figure 3a) and fluorescence microscopy (Figure 3b) analyses of HEK-293T cells overexpressing both HDAC6 (N-terminal HA-tagged wild-type (wt)-HDAC6 (HA-wt-HDAC6) [100,101,106] or wt-HDAC6-DsRed [100]) and NS5-myc constructs showed that functional tubulin-deacetylase HDAC6 targets NS5-myc (Figure 3a, quantified in the associated histogram, and Figure 3b). Notably, overexpression of the NS5 construct stabilized the acetylated MTs (Figure 3a, see Ac-tub in lane 2 compared with control lane 1). However, when functional HDAC6 was overexpressed together with NS5, we observed a significant decrease in the expression of the ZIKV NS5 protein (Figure 3a,b) concomitant with a decrease in the acetylation level of the α-tubulin subunit of MTs (Figure 3a, see NS5 (quantified in associated histograms) and Ac-tub in lane 4 compared with lane 2). Cells overexpressing functional HDAC6 lost the nuclear pattern of ZIKV NS5 protein expression (Figure 3b), and the NS5 protein was not detected in cells overexpressing HDAC6 by fluorescence microscopy.

To determine the degradative pathway used by HDAC6 to target the ZIKV NS5 protein, we inhibited two key functions of HDAC6 involved in its autophagy antiviral action, the deacetylase activity, which resides in two functional deacetylase domains [136,137,138,139,140,141], and the BUZ domain [91,94,100,101,106]. As in the experiment presented in Figure 2b, we first inhibited the deacetylase domains of endogenous HDAC6 by using a specific chemical inhibitor of the enzyme tubacin (Figure 3c). Remarkably, in cells overexpressing NS5-myc, tubacin-mediated inhibition of the endogenous tubulin-deacetylase HDAC6 allows the ZIKV NS5 protein to be expressed not only in the nucleus (as observed for the different nuclear NS5 structures in the control nontreated cells) (Figure 3c) but also widely in the cytoplasm (Figure 3c). Moreover, we analysed the effect exerted on the ZIKV NS5 protein by the well-characterized HDAC6 double deacetylase-dead mutant (dm) (N-terminal HA-tagged (HA-dm-HDAC6)), which is not able to deacetylate HDAC6 substrates and negatively affects the autophagy function of the enzyme [100,101,106]. Thus, biochemical analysis of cells overexpressing both the ZIKV NS5 protein and the dm-HDAC6 mutant revealed that the NS5 protein is more stable and more than six-fold more highly expressed in these cells (Figure 3d, lane 6 (quantified in the associated histograms)) than in cells expressing only the ZIKV NS5 protein (Figure 3d, lane 2 (quantified in the associated histograms)), suggesting that the catalytic activity of HDAC6 is key for the degradation of the ZIKV NS5 protein. This dm-HDAC6-associated “protective” effect on the NS5 protein could be explained by the competitive action exerted by the overexpressed dm-HDAC6 mutant on the endogenous HDAC6 that avoids HDAC6-mediated NS5 degradation. Therefore, these data confirmed that endogenous HDAC6 restricts ZIKV NS5 protein expression and is preserved from the antiviral enzyme in the nucleus (Figure 1b,c, Figure 2a,b and Figure 3b,c (control conditions)).

Once again, functional wt-HDAC6 promoted the clearance of the ZIKV NS5 protein (Figure 3d, lane 5 (quantified in the associated histograms)) concomitantly with the deacetylation of MTs (Figure 3d, lane 5). Thus, the ZIKV NS5 protein could not promote the acetylation of MTs in cells overexpressing functional HDAC6 (Figure 3d, lane 5). Taken together, these data indicate that HDAC6 controls the level of expression and the cellular distribution of the ZIKV NS5 protein and requires its deacetylase activity, which is key for the autophagy function of HDAC6 [100,101].

A well-characterized HDAC6 construct lacking the C-terminal BUZ domain (HA-HDAC6-DBUZ), which is involved in autophagy activation and its degradative activity [100,101,106], was not able to promote clearance of the ZIKV NS5 protein (Figure 4a, comparing lanes 2 and 4 (quantified in the NS5 histogram)). This HA-HDAC6-DBUZ construct could deacetylate MTs (Figure 4a, comparing lanes 3 and 4 with lanes 1 and 2) and promote the stabilization of the autophagy marker p62 (Figure 4a, comparing lanes 3 and 2 with control lane 1) due to the absence of the BUZ region of the enzyme, as we reported [100,101]. Notably, the ZIKV NS5 protein negatively alters p62-autophagy flux, as observed by the accumulation of p62 in cells overexpressing NS5 compared with that in control cells (Figure 4a, comparing lanes 2 and 1) and in cells coexpressing the HA-HDAC6-DBUZ construct (Figure 4a, comparing lanes 2 and 4 with control lane 1). In fact, the expression of the ZIKV NS5 protein, the HDAC6-DBUZ mutant or both constructs increased the expression of the p62 protein (Figure 4a, lanes 2–4, quantified in the associated p62 histogram) compared to the level detected in control cells (Figure 4a, control lane 1, quantified in the associated p62 histogram). Therefore, experimental conditions in cells overexpressing NS5, HA-HDAC6-ΔBUZ or both constructs are aligned with the impairment of the p62-autophagy flux, which should correspond to higher levels of expression of the p62 protein (Figure 4a, lanes 2–4) compared with control cells (Figure 4a, lane 1). These observations indicate that the ZIKV NS5 protein could negatively influence autophagic p62 flux to protect itself from fading and accomplish its viral functions. In fact, the interplay between autophagy and different ZIKV proteins in the context of virus infection and transmission is a very complex subject of research (reviewed in [142,143]).

We further analysed the effect of the chemical inhibition of the autophagy-associated process exerted by 3-methyladenine (3-MA) on HDAC6-mediated ZIKV NS5 degradation in HEK-293T cells. The 3-MA inhibitor was used as previously reported [101] to inhibit the formation of aggresomes and the associated HDAC6-triggered autophagic degradation of targeted proteins [100,101,144,145,146]. 3-MA inhibits the autophagic sequestration of cytoplasmic proteins, as it was previously reported to block HDAC6-mediated autophagic degradation of the HIV-1 Vif and Pr55^Gag^ proteins [98,99,100,101].

Thus, in 3-MA-treated cells, the blockade of the autophagic process is signalled by the stabilization of the NS5 protein in the presence of overexpressed functional HDAC6 compared with that in the control conditions (PBS vehicle control), where NS5 is faded by the HDAC6 degradative effect (Figure 4c, 3-MA vs. PBS NS5 bands in associated Western blot lanes 4, compared with their associated control lane 2; quantified in associated histograms). Furthermore, the ability of HDAC6 to degrade the ZIKV NS5 protein was validated in cells in which endogenous HDAC6 was specifically silenced by short interfering RNA (siRNA). HDAC6 knockdown provoked a significant increase in the expression of the ZIKV NS5 protein compared with that in control, scrambled-treated cells (Figure 4d, HDAC6 silencing and concomitant NS5 stabilization are quantified in the associated histograms).

Therefore, these data suggest that HDAC6 controls the expression of ZIKV NS5 by targeting the viral protein via autophagy, which requires both its deacetylase activity and the BUZ domain of HDAC6.

### 3.4. Autophagy Controls the Stability of the ZIKV NS5 Protein and Its Regulatory Immune Activity on RNA-Mediated RIG-I IFN Production

It has been reported that NS5 allows ZIKV evasion of IFN-mediated innate immunity [61,62,66,68,71,82,147,148,149,150,151]. The ZIKV NS5 protein targets each level of the IFN activation axis in host cells, particularly impairing genomic RNA sensing at its 5′ untranslated region (UTR), which is recognized by retinoic acid-inducible gene I (RIG-I); a cytosolic pattern recognition receptor (PRR) that mediates the induction of a type-I IFN (IFN1) response [152,153,154]. In fact, NS5 represses RIG-I K63-linked polyubiquitination by means of the NS5-MTase (methyltransferase) domain, but MTase functions independently, thereby preventing RIG-I from activating IRF3 and therefore IFN-β production [63] or by acting as a barrier to IFN activation (reviewed in [82]).

Furthermore, we analysed the effect of the ZIKV NS5 protein on RNA-mediated RIG-I IFN production and its ability to inhibit IFN production in cells in which autophagy-mediated NS5 clearance was abrogated. For this purpose, we could not use either a tubacin inhibitor or HDAC6-overexpressing mutants altered in the autophagic-degradative functions of the antiviral enzyme since RIG-I is a substrate for HDAC6 [155,156,157]. Thus, specific chemical inhibition or mutation of the deacetylase domains of HDAC6, as well as the use of the HDAC6-DBUZ mutant, which is still deacetylase active, could alter the functional status of RIG-I, which directly depends on its acetylation state [155,156,158]. Moreover, as we are characterizing here, the degradative effect exerted by HDAC6 on the NS5 protein through its functions (i.e., deacetylases and BUZ-HDAC6 domains) could directly affect NS5 protein expression and the associated IFN interference. Hence, we assayed 3-MA in a HEK-based cell model for IFN production (HEK-Lucia^TM^ RIG-I cells; see materials and methods) to inhibit the formation of aggresomes and the associated HDAC6-triggered autophagic degradation of targeted proteins, as previously reported [100,101,144,145,146].

We observed that the ZIKV NS5 protein inhibited RNA-mediated RIG-I IFN production in HEK-Lucia™ RIG-I cells (Figure 5a, compared with cells not expressing NS5), a model for measuring IFN production in a 5′ triphosphate hairpin RNA (3p-hpRNA) RIG-I agonist-dependent manner [159,160,161,162]. In the presence of 3-MA, NS5 strongly diminished the IFN response mediated by the RIG-I agonist 3p-hpRNA (Figure 5a). The treatment of cells not expressing NS5 with 3-MA decreased the capacity of the RIG-I agonist 3p-hpRNA to promote IFN production. This effect could be a consequence of the complex interplay between autophagy and RIG-I/IFN responses (reviewed in [163]). Likewise, the IFN response mediated by the RIG-I agonist 3p-hpRNA was inhibited by the overexpression of NS5 in 3-MA-treated cells, correlating this neutralizing effect of the ZIKV NS5 protein on IFN production (Figure 5b) with the stability of the NS5 protein after autophagy inhibition (Figure 5b, compare NS5 bands in Western blot lanes 4 and 3). Similarly, biochemical analysis of NS5-expressing HEK-Lucia™ RIG-I cells revealed the stabilization of MTs, as indicated by the increase in the acetylation of the α-tubulin subunits of MTs (Figure 5b, Ac-tub bands were compared between lanes 3 and 4 and lanes 1 and 2).

Taken together, these data suggest that HDAC6-mediated autophagy targeting the ZIKV NS5 protein regulates NS5 cell distribution and localization and that the associated aberrant effects on the tubulin cytoskeleton control the ability of NS5 to interfere with the IFN response mediated by the RIG-I pathway activated by foreign RNA agonists.

## 4. Discussion

The ZIKV NS5 protein is required for virus genome replication, and one of the ZIKV proteins involved in immune evasion of the IFN-mediated innate response [61,62,63,64,65,66,67,68,69,70,71,72,73,74,75,76,77,78,79,80,81,82]. In this work, we aimed to study potential cell structures and functions that could be affected by the ZIKV NS5 protein and considered potential mechanisms for cell toxicity and pathogenesis associated with ZIKV infection. Our results obtained in HEK-293T cells, a model for ZIKV and other RNA virus infection and production [98,99,100,101,125,126,127], suggested that the ZIKV NS5 protein could aberrantly reorganize the tubulin cytoskeleton by promoting nested structures of acetylated MTs organized at the periphery of the cell. Although the caged reorganization of MTs has been reported in ZIKV-infected cells [88], which surround the viral replication factory close to the endoplasmic reticulum (ER), we believe that this is the first time that this caged effect on MTs has been reported to be associated with the ZIKV NS5 protein. These kinds of nested structures have also been described for Bunyamwera virus (BUNV) but for virus-triggered aberrant actin cytoskeleton reorganization [164]. Notably, BUNV belongs to another family of emergent RNA enveloped viruses that can cause severe episodes of encephalitis and haemorrhagic fevers in humans [165,166,167,168,169]. It is important to consider that the use of an infectious ZIKV avoids establishing a clear-cutting association of the rearrangements observed on MTs with a particular ZIKV protein, as previously reported [88]. The authors did not associate the effects observed on the cytoskeleton with any viral factor, such as the NS5 protein. A virion lacking NS5 is required to determine NS5 function together with the chance that no other ZIKV protein or viral factor contributes to the observed effects. Due to the importance of the ZIKV NS5 protein for viral replication [66,170,171,172], it would be difficult to produce and expand delta-NS5 virions, and deleterious mutations in NS5 could abrogate ZIKV replication [173,174,175] or enhance its toxicity (i.e., NS5 2634 and 3328 mutations) [176], as it is not easy to obtain viral progeny (i.e., this virus could be toxic to bacteria and packaging cells) [176]. It is possible to nullify the toxic effect of NS5 in the bacteria used for the development of the infectious cDNA clone of ZIKV by sequence modification (i.e., inserting a second copy of the intron sequence after nucleotide position 8882 in NS5) [177], but this procedure does not delete NS5 from virions. For this reason, we performed experiments with a recombinant ZIKV NS5 protein to determine the functional effects of the viral protein on cells.

The ZIKV NS5 protein appears to be located mainly in nuclear structures, as previously reported [64,65,117,118], according to fluorescence microscopy analysis. This observation was confirmed by biochemical analysis of cell fractions from cells overexpressing the ZIKV NS5 protein, which was mainly distributed in the nuclear fraction and was only slightly detected in the cytoplasmic fraction. Some studies have associated the ZIKV NS5 protein with the MTOC or centrosome during mitosis and cell division [95,114,115,116]. It is conceivable that the cytoplasmic pool of the ZIKV NS5 protein could alter MT dynamics and stability. However, we cannot rule out that the ZIK NS5 protein must drive MT reorganization from the nucleus, as reported for nuclear elements that can establish transient or stable connections with MTs (reviewed in [178]), where aberrant crosstalk between MTs and the nucleus may contribute to a plethora of human developmental defects and other diseases [179,180,181,182,183]. Similarly, abnormal tubulin cytoskeleton organization and functions could be responsible for a plethora of named primary tubulinopathies and neurodegenerative diseases [184]. As described in the introductory section of this work, a hallmark of ZIKV infection is CZS (i.e., including microcephaly) together with abnormal brain development and other neurologic manifestations [27,41,42,43,44,45] and disorders [34,46,47,48,49], such as GBS [46,48,50,51,52], encephalitis [46,51,52,53], meningoencephalitis [34,54], acute myelitis [46,52,55] and encephalomyelitis [46,49,51,56,57]; as well as sensory polyneuropathy [58] and other neurological complications [59,60]. Similarly, it has been recently reported that the ZIKV NS5 protein interacts with components of the cilium base, promoting ciliopathy and premature neurogenesis [96]. Therefore, the NS5-induced nested rearrangement of MTs at the periphery of the cell could be considered a cell anomaly that, in part, could be responsible for cell toxicity, abnormal tissue development and/or function and ZIKV infection pathology.

Although it is a unique flavivirus that impacts cellular architecture, it is also exceptional in that it controls and restricts its NS5 protein in nuclear discrete bodies, as we described here and has been previously reported [64,65,117,118]. This peculiar way of organizing could mechanistically explain how the virus supports infection [88]. In fact, ZIKV seems to require MTs and the MTOC for viroplasm organization and virus production [95]. Remarkably, the authors observed that viroplasm formation and ZIKV production are not negatively affected when infected cells have no centrosomes or an impaired MTOC, suggesting that viroplasm acts as a potential site for MT organization and efficient virus production [95]. It is well established that MTs are commonly hijacked by viruses to traffic to sites of replication after entry and to promote egress from infected cells (reviewed in [92,111]). Therefore, the NS5 protein could assure efficient viral production and egress by organizing nested MT platforms located at the periphery of the cells, as we observed.

In this regard, one of the main cellular functions that regulates viral infection and is closely related to the organization of MTs is the autophagy process [89,90,91,92,93,94]. Although the complexity of the interplay between autophagy and different ZIKV proteins in the context of virus infection and transmission has been reported [142,143,185,186], little is known about the ZIKV NS5 protein. Our observations indicate that the ZIKV NS5 protein negatively alters p62-autophagy flux in a dose-dependent manner, as indicated by the accumulation of p62 in cells overexpressing NS5. Therefore, the ZIKV NS5 protein affects cell structures and events that are key for cell permissivity to viral infection and survival, such as tubulin cytoskeleton dynamics or p62-associated autophagy flux [97,98,99,100,101,102,103], which are functions regulated by the antiviral factor HDAC6 [91,94,98,99,100,101,104,105,106,107,108], presumably to protect itself from fading and accomplishing its viral functions. Remarkably, p62 and HDAC6 are associated with aggresomes and the autophagy-derived pathway [97,102,122,187,188], which has been reported to act against ZIKV infection [189]. In this study, the authors compete for HDAC6-mediated aggresome formation by using small synthetic “designed ankyrin repeat proteins” (DARPins) that impair the interaction between HDAC6 and ubiquitin. This DARPins-associated action blocked infection by ZIKV and the influenza A virus (IAV) at the uncoating step of the viral cycle [189].

Taken together, our results indicate that HDAC6 targets the NS5 protein via autophagy in a deacetylase- and BUZ domain-dependent manner and controls the cytoplasmic expression of the ZIKV NS5 protein. Similarly, by using HDAC6 mutants lacking key functions responsible for the autophagy activity of antiviral enzymes, such as deacetylases and BUZ domains, the results confirmed that the ZIKV NS5 protein is under the degradative control of HDAC6. Functional HDAC6 targets NS5 by autophagy, as the degradative effect of HDAC6 on NS5 is blocked by the 3-MA aggresome/autophagy inhibitor or by the overexpression of the deacetylase dead-mutant (dm-HDAC6) or DBUZ mutant (HDAC-DBUZ) of HDAC6. The siRNA-mediated knockdown of endogenous HDAC6 indicated that the endogenous enzyme targeted the ZIKV NS5 protein. Remarkably, fluorescence microscopy revealed that the ZIKV NS5 protein is present both in the nucleus and in the cytoplasm of cells when the deacetylase activity of HDAC6 is specifically inhibited by tubacin. Notably, tubacin-mediated inhibition of HDAC6 deacetylase activity induces MT acetylation but does not cause the caged reorganization of MTs observed for the ZIKV NS5 protein. Therefore, it is conceivable that the effect of NS5 on aberrant MT rearrangement does not involve an inhibitory effect of the viral protein on the tubulin deacetylase activity of endogenous HDAC6. Taken together, these data suggest that functional HDAC6 negatively regulates the protein expression of NS5 in the cytoplasm, which is predominantly detected in the nucleus, and prevents the aberrant acetylation of MTs and p62 accumulation observed when the ZIKV NS5 protein is overexpressed. Thus, it is plausible that, as occurs with HIV, IAV and other viruses [91,92,94,98,99,100,101,104,105,106,190,191], HDAC6 could also be involved in the control of ZIKV NS5-driven viral RNA replication and virus production at the late stages of the viral cycle and that the ZIKV NS5 protein targets the p62/HDAC6 aggresome pathway to ensure viral persistence.

Moreover, our results indicate that the ZIKV NS5 protein inhibits RNA-mediated RIG-I IFN production in HEK-Lucia RIG-I-adapted cells and that the inhibitory activity of the viral protein increases when autophagy is inhibited by the chemical inhibitor 3-MA. This inhibits the formation of aggresomes and the associated HDAC6-triggered autophagic degradation of targeted proteins [100,101,144,145,146]. In fact, 3-MA impairs autophagic sequestration and HDAC6-mediated degradation of the HIV-1 Vif and Pr55^Gag^ proteins [98,99,100,101]. Thus, the results obtained with 3-MA for NS5 inhibition of RIG-I-mediated IFN production, triggered by the RIG-I agonist 3p-hpRNA (5′ triphosphate hairpin RNAs), are in accordance with the increase observed in the level of ZIKV NS5 protein expression when endogenous HDAC6-associated autophagy on NS5 is blocked by 3-MA.

Notably, HDAC6 mediates the deacetylation of the C-terminal region of RIG-I, thereby promoting the viral RNA-sensing activity of RIG-I and the associated IFN response [155]. Therefore, similar to the acetylation of MTs (a substrate of HDAC6) promoted by NS5, it is conceivable that the ZIKV NS5 protein may act on the RIG-I substrate of HDAC6, affecting its deacetylation state and activity to inhibit IFN production. The ZIKV NS5 protein targets each level of the IFN activation axis in host cells, particularly impairing genomic RNA sensing at its 5′ untranslated region (UTR) capped by RIG-I by repressing RIG-I polyubiquitination by means of the NS5-MTase (methyltransferase) domain but independent of MTase function, thereby preventing RIG-I from activating IFN regulatory factor 3 (IRF3) and therefore IFN-β production [63] or by acting as a barrier to IFN activation. Hence, NS5 abrogates IRF3 and nuclear factor kappa B (NF-κB) signalling [71], as it can interact with IRF3, preventing its activation [192], or bind to inhibitor of kappa-B kinase epsilon (IKKε), affecting IKKε protein levels and phosphorylation, which ultimately results in IRF3 inactivation [151]. Furthermore, the NS5 protein antagonizes IFN production by impairing the activation of TANK-binding kinase 1 (TBK1) and, therefore, IRF3, a TBK1 substrate for phosphorylation [61]. In addition, NS5 competitively binds to the ubiquitin-like domain of TBK1, affecting its interaction with tumour necrosis factor (TNF)-associated factor 6 (TRAF6). This complex is required for TBK1-mediated IRF3 phosphorylation and activation, thus interfering with type I and III IFN transcription [61]. Furthermore, in IFN-induced human cells, ZIKV NS5 expression results in proteasomal degradation of the IFN-regulated transcriptional activator STAT2 (signal transducer and activator of transcription 2) in humans and affects STAT1 phosphorylation levels, suppressing INF-mediated Janus kinase (JAK)/STAT signal transduction [62,149], which shows that ZIKV-induced disease takes advantage of NS5-promoted IFN deficiency. Moreover, although the role of NS5 in the nucleus remains enigmatic, recent studies have suggested that the subcellular localization of NS5 is important for its function in innate immune suppression, which provides new insight into ZIKV pathogenesis [65,118,193]. Thus, the inhibition of the RIG-I/IFN response by ZIKV NS5 is thought to be key to ensuring the viral cycle and the infection of cells found in immunoprivileged sites, such as the brain or placenta [71,194], thereby allowing ZIKV evasion of IFN-mediated innate immunity. Therefore, the RIG-I/HDAC6 interplay could be key for the control of the IFN-mediated immune response against ZIKV by acting on NS5.

## 5. Conclusions

Taken together, the above results suggest that the ZIKV NS5 protein contributes to cell toxicity and pathogenesis by affecting MT dynamics and p62-associated autophagic flux and evades the IFN-immune response by overcoming HDAC6 functions. Moreover, through autophagy targeting of the ZIKV NS5 protein, HDAC6 has emerged as a key anti-ZIKV factor that can control NS5-mediated functions in cells, such as aberrant alteration of the tubulin cytoskeleton and inhibition of autophagic p62 flux; thus, HDAC6 is a potential protective factor against cell toxicity and associated ZIKV infection pathogenesis.

## Figures and Tables

**Figure 1 cells-13-00598-f001:**
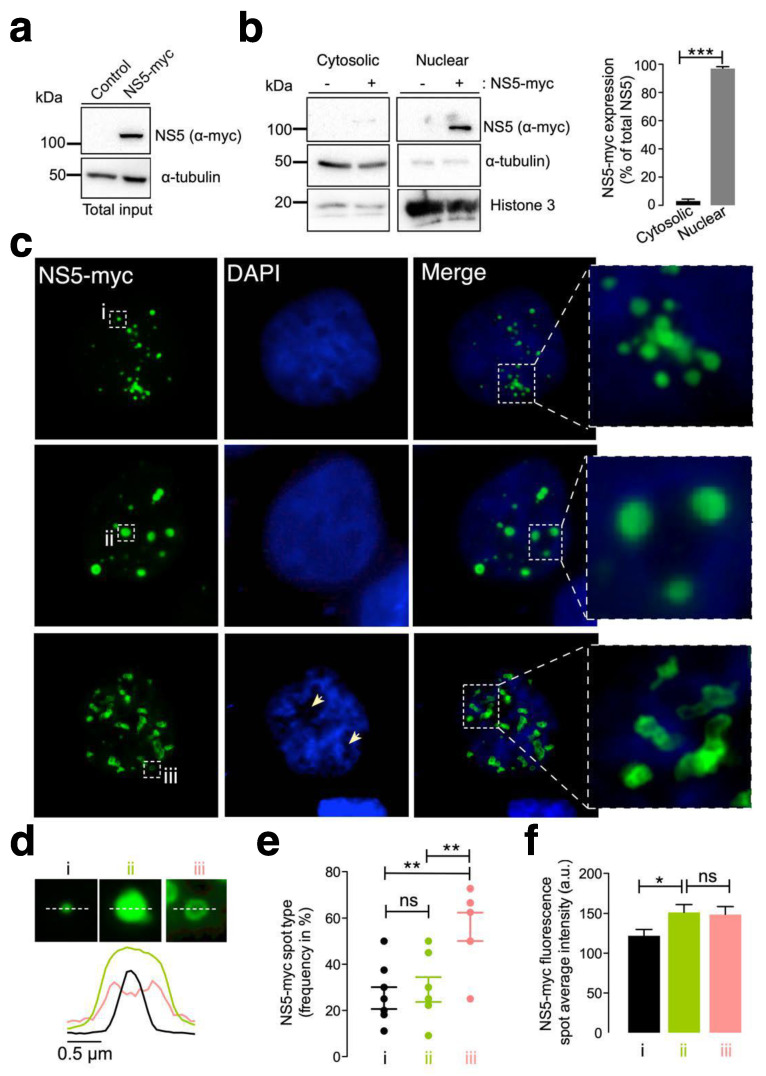
Characterization of the ZIKV NS5 protein expression pattern. (**a**) Quantitative Western blot analysis of the C-terminal myc-tagged NS5 (NS5-myc) viral protein construct (1.5 µg cDNA) and total α-tubulin protein in a viral packaging HEK-293T cell model compared to those in control cells transfected with the pcDNA^TM^ 3.1(+) plasmid (1.5 µg cDNA). The α-tubulin protein was used as the control for total protein loading. A representative of two experiments is shown (see associated data in Appendix A). (**b**) Left, Quantitative Western blot analysis of the cell fraction of HEK-293T cells overexpressing NS5-myc (1.5 µg cDNA) and its distribution. Histone 3 was used as the protein control for the nuclear fraction, whereas α-tubulin was used for the cytoplasmic fraction and total protein load. A representative of two experiments is shown (see associated data in Appendix A). Right, histograms quantifying the distribution of nuclear and cytoplasmic NS5-myc in lysates from (**b**) experiments, normalized to total α-tubulin. NS5 is mostly found in the nucleus. The data are presented as the means ± S.E.M. of two independent experiments. (**c**) Immunofluorescence microscopy images of HEK-293T cells overexpressing NS5-myc (1.5 µg cDNA) and fixed at 48 h post transfection. The subcellular localization of NS5 was determined using a specific α-myc mAb (green). DAPI is a blue, fluorescent probe for DNA staining. Images were captured using a 100× objective and analysed with the MetaMorph program. NS5 is found in the nucleus and forms three types of structure: uniform punctate shapes ranging from one (type i) to several (type ii) micrometres in diameter, and spheroidal and ring-shaped shapes with longer sizes (type iii). Arrowheads denote the lack of DNA and NS5 colocalization. (**d**) Line scanning of the different structures containing the NS5 viral protein (α-myc mAb (green)) (type i, black lines; type ii, green lines; and type iii, pink lines). (**e**) Quantification of the frequency of the presence of different NS5 structures (type i, black dots; type ii, green dots; and type iii, pink dots) per cell and statistical analysis (*n* = 80 cells). (**f**) Average fluorescence intensity plot of NS5 spots in different structures (type i, black bar; type ii, green bar; and type iii, pink bar) per cell (*n* = 80 cells). In this figure, when indicated, the *p* values are *** *p* ≤ 0.001, ** *p* ≤ 0.01 and * *p* ≤ 0.05; ns indicates not statistically significant (*p* > 0.05). The *p* value is the comparison of the means between the two groups using the parametric Student’s *t* test.

**Figure 2 cells-13-00598-f002:**
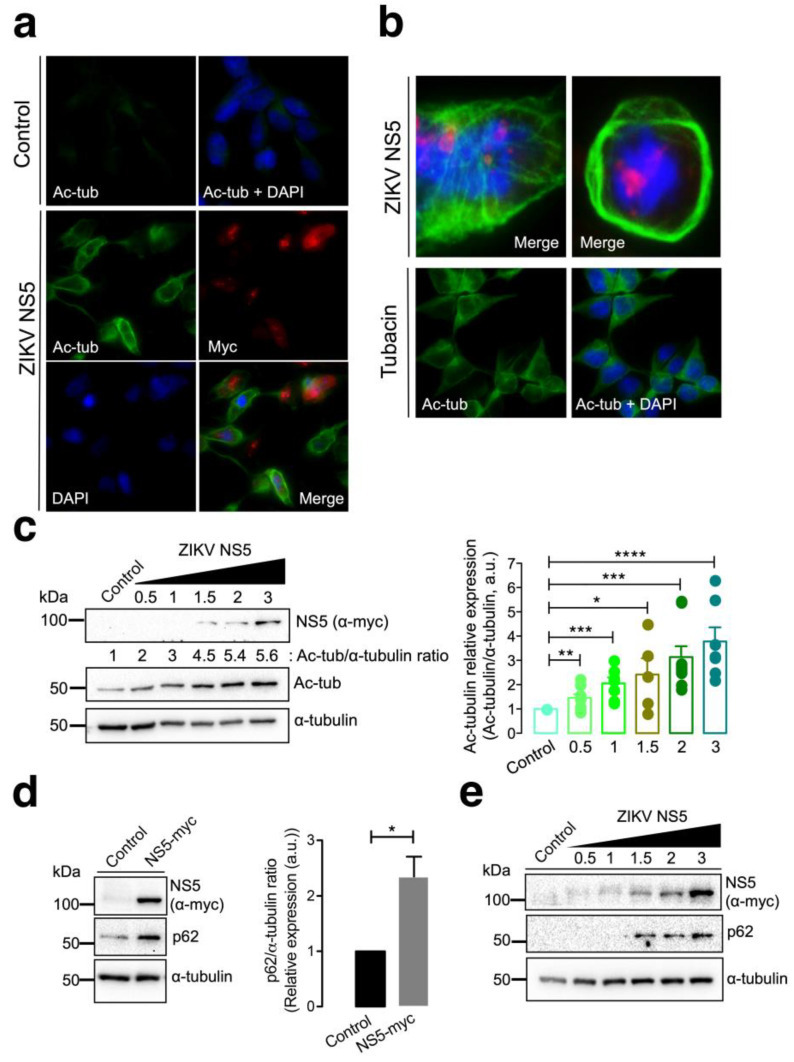
Characterization of the effect of the ZIKV NS5 protein on microtubule stabilization and reorganization and on the accumulation of the autophagy marker p62. (**a**) Immunofluorescence microscopy analysis of the effect of NS5 on microtubules (MTs) in HEK-293T cells overexpressing NS5-myc (1.5 µg cDNA) compared to that in control cells transfected with the pcDNA^TM^ 3.1(+) plasmid (1.5 µg cDNA). In this representative experiment, cells were processed for immunostaining with antibodies against the acetylated Lys^40^ residue in α-tubulin (green) and/or NS5 (using an anti-myc mAb) (red). DAPI is a blue, fluorescent probe for DNA staining. Images were captured using a 40× objective and analysed with the MetaMorph program. (**b**) Immunofluorescence microscopy images showing the exacerbated nested structure of acetylated MTs promoted in cells overexpressing NS5 (1.5 µg cDNA) (images captured using a 100× objective). As a control for the acetylation of MTs, cells were treated with tubacin (1 μM), a specific inhibitor of the deacetylase activity of HDAC6. This image (captured using a 40× objective) also serves as a control for the NS5-mediated acetylation of MTs images presented in panel (**a**). In these representative experiments, cells were processed for immunostaining with antibodies against the acetylated Lys^40^ residue in α-tubulin (green) and/or NS5 (using an anti-myc mAb) (red). DAPI is a blue, fluorescent probe for DNA staining. Images were captured using a 100× or 40× objective, as indicated, and analysed with the MetaMorph program. (**c**) Quantitative Western blot analysis of the dose–response effect of NS5 overexpression on MT stabilization, as monitored by the acetylation of MTs at Lys^40^ α-tubulin residues. The α-tubulin protein was used as the loading control protein. A representative of nine independent experiments is shown (see associated data in Appendix A). Histograms quantifying the amount of acetylated α-tubulin induced by NS5 in stabilized MTs from nine replicates of the experiment shown in panel (**c**). The data were normalized to the total α-tubulin protein. (**d**) Western blot analysis of the effect of the NS5 viral protein (1.5 µg cDNA) on the accumulation of the autophagic protein p62 in HEK-293T cells. NS5, p62 and total α-tubulin are shown. Histograms quantifying the amount of p62 stabilized by NS5 from the experiments are shown in panel (**d**). The data were normalized to the total α-tubulin protein. The data are presented as the means ± S.E.M. of three independent experiments (see associated data in Appendix A). (**e**) Quantitative Western blot analysis of the dose–response effect of NS5 overexpression on the accumulation of the autophagic protein p62 in HEK-293T cells overexpressing NS5. NS5, p62 and total α-tubulin are shown. The data are presented as the means ± S.E.M. of four independent experiments (see associated data in Appendix A). In panels (**c**–**e**), when indicated, the *p* values are **** *p* ≤ 0.0001, *** *p* ≤ 0.001, ** *p* ≤ 0.01 and * *p* ≤ 0.1; ns indicates not statistically significant (*p* > 0.05). The *p* value is the comparison of the means between the two groups using the parametric Student’s *t* test.

**Figure 3 cells-13-00598-f003:**
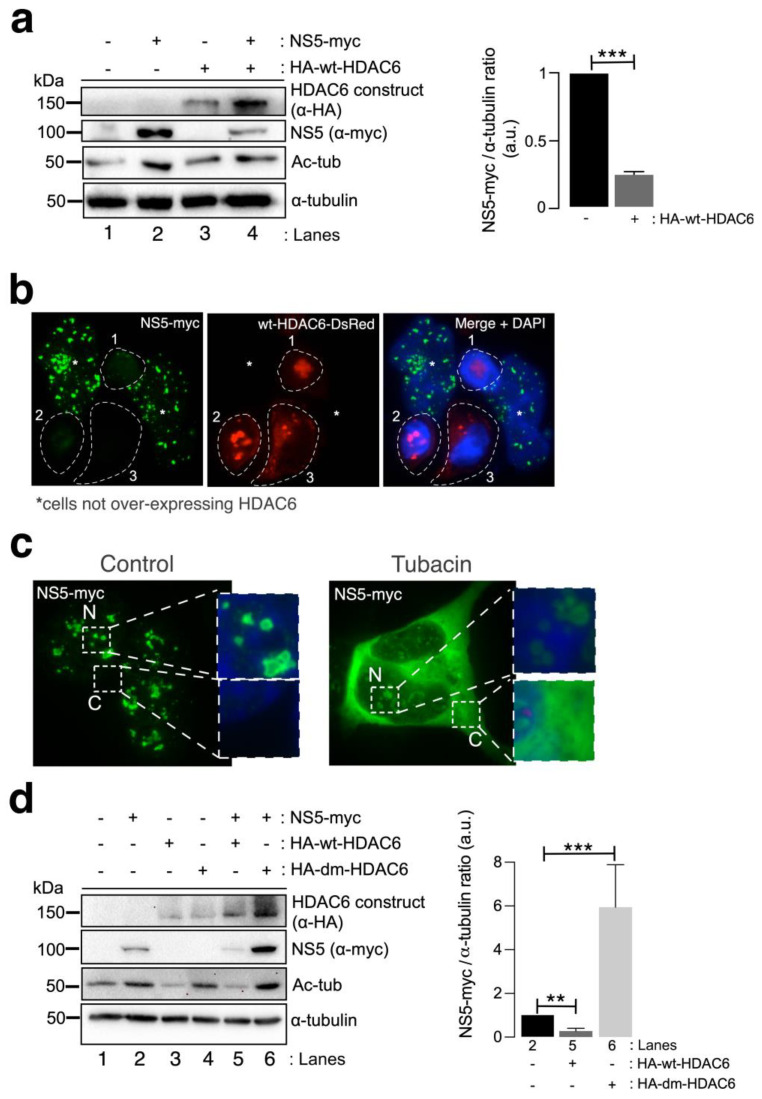
HDAC6 targets the ZIKV NS5 protein in a deacetylase-dependent manner, regulating the expression of NS5 in the cytoplasm. (**a**) Quantitative Western blot analysis of HEK-293T cells overexpressing NS5-myc (lane 2), N-terminal HA-tagged wt-HDAC6 (HA-wt-HDAC6) (lane 3) or both constructs (lane 4) (1.5 µg cDNA) and comparison to control cells transfected with the pcDNA^TM^ 3.1(+) plasmid (1.5 µg cDNA) (lane 1). The negative effects of HA-wt-HDAC6 on NS5-myc expression (lane 4), the stabilization of acetylated MTs by NS5 (lane 2) and the deacetylase activity of HDAC6 on acetylated MTs (lanes 3 and 4) are shown. The α-tubulin protein was used as the loading control protein. A representative experiment of four is shown (see associated data in Appendix A). Histograms quantifying the amount of NS5-myc expressed in the absence (corresponding to lane 2 in the left panel) or presence of overexpressed HDAC6 (corresponding to lane 4 in the left panel) normalized to total α-tubulin. The data are presented as the means ± S.E.M. of four independent experiments. (**b**) Immunofluorescence images showing NS5 labelling 48 h post transfection in cells cotransfected with HDAC6-DsRed (red), fixed and labelled with an α-myc antibody for NS5 detection (green). NS5 was detected only in cells not overexpressing wt-HDAC6-DsRed (white asterisks). The white dotted lanes indicate the cell perimeters of three cells (1–3) that expressed wt-HDAC6-DsRed in the field. DAPI is a blue, fluorescent probe for DNA staining. Images were captured using a 100× objective and analysed with the MetaMorph program. (**c**) Immunofluorescence microscopy analysis of NS5 cell distribution in cells treated or not treated (control) with tubacin (1 μM), a deacetylase inhibitor of HDAC6, in cells overexpressing NS5-myc (1.5 µg cDNA). The cells were processed for immunostaining with an anti-myc mAb (green). DAPI is a blue, fluorescent probe for DNA staining. Images were captured using a 100× objective and analysed with the MetaMorph program. (**d**) Quantitative Western blot analysis of the effect of functional (HA-wt-HDAC6) or deacetylase inactive (HA-dm-HDAC6) HDAC6 on the stability of the ZIKV NS5 protein (NS5-myc) in HEK-293T cells overexpressing the different constructs (1.5 µg cDNA per construct) and comparison with cells overexpressing either only NS5 or only HA-wt-HDAC6 or HA-dm-HDAC6. The levels of acetylated α-tubulin and total α-tubulin under the different experimental conditions are shown. A representative experiment of two is shown (see associated data in Appendix A). Histograms quantifying the amount of NS5 detected in cells from the experiment (**d**). The data are presented as the means ± S.E.M. of two independent experiments. For the quantitative analysis of the Western blots, when indicated, the *p* values are *** *p* ≤ 0.001 and ** *p* ≤ 0.01. The *p* value is the comparison of the means between the two groups using the parametric Student’s *t* test.

**Figure 4 cells-13-00598-f004:**
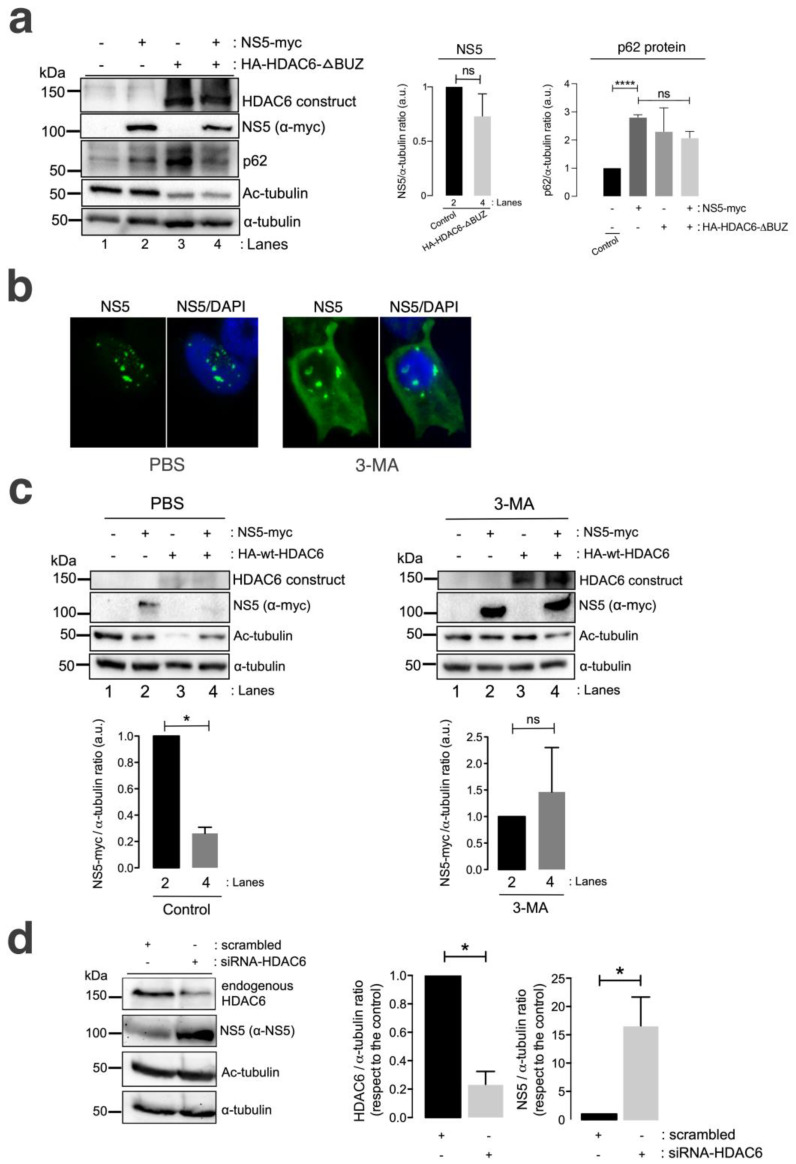
HDAC6 promotes the autophagic clearance of the ZIKV NS5 protein. (**a**) Quantitative Western blot analysis of HEK-293T cells overexpressing NS5-myc (lane 2) or the N-terminal HA-tagged HDAC6 mutant lacking the C-terminal BUZ domain (HA-HDAC6-DBUZ) (lane 3) or both constructs (lane 4) (1.5 µg cDNA) and comparison to control cells transfected with the pcDNA^TM^ 3.1(+) plasmid (1.5 µg cDNA) (lane 1). The absence of a degradative effect of HA-HDAC6-DBUZ on NS5-myc expression (lane 4) and the deacetylation of MTs by HA-HDAC6-DBUZ, which is deacetylase active (lanes 3 and 4) are shown. NS5 could not stabilize acetylated MTs under these experimental conditions (lane 4). The α-tubulin protein was used as the loading control protein. A representative experiment of three is shown (see associated data in Appendix A). Left histograms quantifying the amount of NS5-myc expressed in the absence (corresponding to lane 2 in the western blot panel) or presence of overexpressed HA-HDAC6-DBUZ (corresponding to lane 4 in the western blot panel), normalized to total α-tubulin. Right histograms quantifying the amount of p62 protein expressed in cells overexpressing NS5-myc, HA-HDAC6-DBUZ or both constructs (corresponding to lanes 2–4 in the western blot panel, respectively) and compared with that in control cells (corresponding to lane 1 in the western blot panel). These data are presented as the means ± S.E.M. of three independent experiments. (**b**) Immunofluorescence microscopy analysis of NS5 cell distribution in cells treated or not treated (control) with 3-MA (5 mM), an inhibitor of HDAC6-mediated aggresome formation and further autophagy clearance of targeted proteins, in cells overexpressing NS5-myc (1.5 µg cDNA). The cells were processed for immunostaining with an anti-myc mAb (green). DAPI is a blue, fluorescent probe for DNA staining. Images were captured using a 100x objective and analysed with the MetaMorph program. (**c**) Quantitative Western blot analysis of HEK-293T cells overexpressing NS5-myc (lane 2), the functional HA-HDAC6 construct (lane 3) or both constructs (lane 4) (1.5 µg cDNA), compared to control cells transfected with the pcDNA^TM^ 3.1(+) plasmid (1.5 µg cDNA) (lane 1) under control conditions (PBS-treated cells, vehicle for 3-MA) or cells treated with 3-MA (5 mM). The absence of a degradative effect exerted by HA-HDAC6 on NS5 in the presence of 3-MA (lane 4 in 3-MA-treated cells compared to lane 4 in PBS-treated cells) and the deacetylation of MTs by HA-HDAC6 are shown. The α-tubulin protein was used as the loading control protein. A representative experiment of 2 is shown (see associated data in Appendix A). Histograms quantifying the amount of NS5-myc expressed in the absence (lane 2) or presence of overexpressed HA-HDAC6 (lane 4), normalized to total α-tubulin, in both PBS- and 3-MA-treated cells are shown. The data are presented as the means ± S.E.M. of 2 independent experiments. (**d**) Quantitative Western blot analysis of the effect of siRNA-mediated knockdown of endogenous HDAC6 (siRNA-HDAC6, 25 pmol/well) on the level of NS5 protein expression in HEK-293T cells overexpressing NS5-myc and compared to that in control and scrambled (1 µM)-treated cells. A representative experiment of four is shown (see associated data in Appendix A). Histograms quantifying the amount of endogenous HDAC6 and NS5 detected in cells treated with siRNA-HDAC6 or scrambled oligos, normalized to total α-tubulin. The data are presented as the means ± S.E.M. of four independent experiments. For quantitative analysis of Western blots, when indicated, the *p* values are **** *p* ≤ 0.0001 and * *p* ≤ 0.1; ns indicates not statistically significant (*p* > 0.05). The *p* value is the comparison of the means between the two groups using the parametric Student’s *t* test.

**Figure 5 cells-13-00598-f005:**
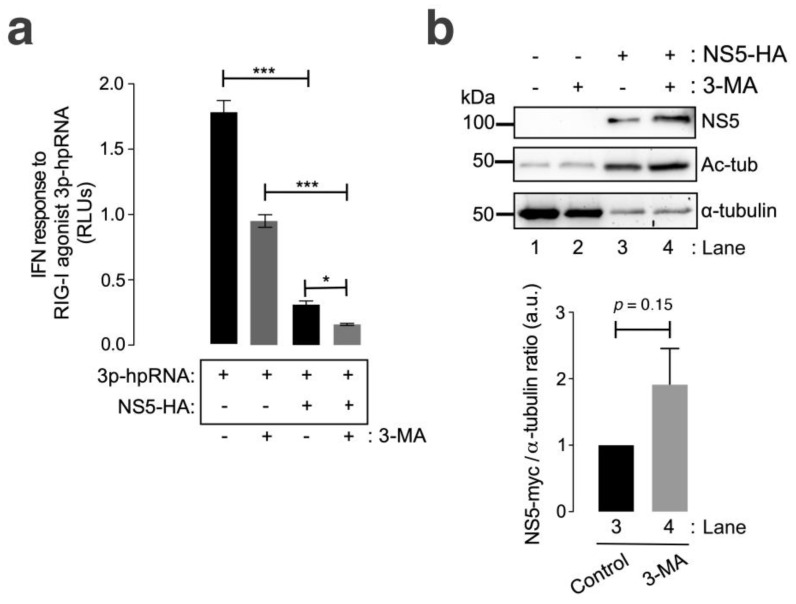
The ZIKV NS5 protein inhibits RNA-triggered RIG-I-mediated IFN production, and this effect is enhanced by autophagy inhibition. (**a**) Quantitative assay of the inhibitory effect of NS5 on RNA-triggered RIG-I-mediated IFN production in HEK-Lucia^TM^ RIG-I cells expressing NS5 (1.5 µg cDNA) and treated with 3p-hpRNA (100 nM), an RNA-specific RIG-I agonist, for 24 h before the IFN-I response was quantified by QUANTI-Luc. The data obtained from control cells transfected with the pcDNA^TM^ 3.1(+) plasmid (1.5 µg cDNA) and treated with 3p-hpRNA (100 nM) are shown. The effect of 3-MA (5 mM) on the NS5-mediated inhibition of 3p-hpRNA-triggered RIG-I-mediated IFN production is shown. The data are presented as the means ± S.E.M. of two independent experiments (*n* = 6). (**b**) Quantitative Western blot analysis of the effect of 3-MA treatment on HEK-Lucia cells overexpressing NS5-myc (1.5 µg cDNA). This biochemical analysis corresponds to the cells shown in panel (**a**) for RNA-mediated RIG-I IFN production. Lanes 1 and 2 represent control cells transfected with the pcDNA^TM^ 3.1(+) plasmid (1.5 µg cDNA) and treated with PBS (vehicle for 3-MA) or 3-MA (5 mM), respectively. Lanes 3 and 4 represent cells overexpressing NS5 (NS5-HA plasmid, 1.5 µg cDNA) and treated with PBS or 3-MA (5 mM), respectively. The stabilization effect of 3-MA on NS5 (lane 4 in 3-MA-treated cells compared to lane 3 in PBS-treated cells) and the NS5-mediated acetylation of MTs are shown. The α-tubulin protein was used as the loading control protein. A representative experiment of two is shown (see associated data in Appendix A). Histograms quantifying the amount of NS5-HA expressed, normalized to total α-tubulin, in both control (PBS)- and 3-MA-treated cells from the top Western blot experiments. The data are presented as the means ± S.E.M. of two independent experiments. In (**a**), the *p* values are *** *p* ≤ 0.001 and * *p* ≤ 0.1, and in (**b**), the *p* value is 0.15. The *p* value is the comparison of the means between the two groups using the parametric Student’s *t* test.

## Data Availability

All presented data are available upon request.

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
