# Peer review of "The ZIKV NS5 Protein Aberrantly Alters the Tubulin Cytoskeleton, Induces the Accumulation of Autophagic p62 and Affects IFN Production: HDAC6 Has Emerged as an Anti-NS5/ZIKV Factor"

_cells, 2024, doi:10.3390/cells13070598_

Round 1
Reviewer 1 Report
Comments and Suggestions for Authors
In the present manuscript authors studied the potential effects of NS5 on microtubules (MTs) and autophagy flux. Authors indicated that NS5 accumulates in nuclear nested-like structures and promotes the acetylation of MTs. They also indicated that NS5 alters events that are under the control of the autophagic tubulin-deacetylase HDAC6.
Moreover, NS5 inhibits RNA-mediated RIG-I interferon (IFN) production, resulting in greater activity when autophagy is inhibited.
The results are interesting. However, the main weakness of the manuscript is that all the experiments were performed with an overexpressed recombinant ZIKV NS5 protein. To confirm the participation of the HDAC6 protein and autophagy and IFN pathway is necessary to perform the experiments with ZIKV infected cells.
Author Response
Reply to Reviewer #1:
Comments and Suggestions for Authors:
In the present manuscript authors studied the potential effects of NS5 on microtubules (MTs) and autophagy flux. Authors indicated that NS5 accumulates in nuclear nested-like structures and promotes the acetylation of MTs. They also indicated that NS5 alters events that are under the control of the autophagic tubulin-deacetylase HDAC6. Moreover, NS5 inhibits RNA-mediated RIG-I interferon (IFN) production, resulting in greater activity when autophagy is inhibited.
We thank Reviewer #1 for this summary and for the comments on our work, which also highlighted the functional role of NS5, which is under the control of HDAC6 and the associated autophagy.
The results are interesting. However, the main weakness of the manuscript is that all the experiments were performed with an overexpressed recombinant ZIKV NS5 protein. To confirm the participation of the HDAC6 protein and autophagy and IFN pathway is necessary to perform the experiments with ZIKV infected cells.
We thank Reviewer #1 for this remark.
As we have stated in the manuscript (former lines 736-740), the caged reorganization of MTs has been reported in ZIKV-infected cells (Ref. [88] by M. Cortese, et al. Cell Rep 2017 Vol. 18 Issue 9 Pages 2113-2123. DOI: 10.1016/j.celrep.2017.02.014), which surround the viral replication factory close to the endoplasmic reticulum (ER). In this work, the use of the virus avoided establishing a clear-cutting association of the rearrangements observed on MTs with a particular ZIKV protein. Likewise, the authors did not associate the effects observed on MTs with any viral factor or, for example, the NS5 protein. A virion lacking NS5 is required to determine NS5 function together with the chance that no other ZIKV protein or viral factor contributes to the observed effects. Due to the importance of the ZIKV NS5 protein for viral replication (Elshahawi H, et al. Importance of Zika Virus NS5 Protein for Viral Replication. Pathogens. 2019 Sep 30;8(4):169. doi: 10.3390/pathogens8040169; Zhao B, et al. Structure and function of the Zika virus full-length NS5 protein. Nat. Commun. 2017;8:14762. doi: 10.1038/ncomms14762), it would be difficult to obtain delta-NS5 virions, and deleterious point mutations in NS5 could abrogate ZIKV replication or enhance its toxicity (i.e., NS5 2634 and 3328 mutations), making it very difficult to obtain viral progeny (i.e., this virus could be toxic to bacteria and packaging cells) (Collette NM, et al. Single Amino Acid Mutations Affect Zika Virus Replication In Vitro and Virulence In Vivo. Viruses. 2020 Nov 12;12(11):1295. doi: 10.3390/v12111295). It is possible to nullify the toxic effect of NS5 by sequence modification (i.e., inserting a second copy of the intron sequence after nt position 8882 in NS5) in the bacteria used for the development of the infectious cDNA clone of ZIKV (Tsetsarkin KA, et al. A Full-Length Infectious cDNA Clone of Zika Virus from the 2015 Epidemic in Brazil as a Genetic Platform for Studies of Virus‒Host Interactions and Vaccine Development. mBio. 2016;7:e01114-16. doi: 10.1128/mBio.01114-16), but this procedure does not delete NS5 from virions.
For these reasons, we performed experiments with a recombinant ZIKV NS5 protein to determine the functional effects of the viral protein on cells. We believe that this is the first time that the caged effect observed on MTs and the increase in the level of expression of the p62 protein are proposed to be exerted by the ZIKV NS5 protein, as well as the characterization of the effect exerted by HDAC6-associated autophagy on NS5 degradation and IFN production.
In this sense, we have added text to the Discussion section (new lines 756-770), as follows:
“It is important to consider that the use of an infectious ZIKV avoids establishing a clear-cutting association of the rearrangements observed on MTs with a particular ZIKV protein, as previously reported [88]. The authors did not associate the effects observed on the cytoskeleton with any viral factor, such as the NS5 protein. A virion lacking NS5 is required to determine NS5 function together with the chance that no other ZIKV protein or viral factor contributes to the observed effects. Due to the importance of the ZIKV NS5 protein for viral replication [66,170-172], it would be difficult to produce and expand delta-NS5 virions, and deleterious mutations in NS5 could abrogate ZIKV replication [173-175] or enhance its toxicity (i.e., NS5 2634 and 3328 mutations) [176], as it is not easy to obtain viral progeny (i.e., this virus could be toxic to bacteria and packaging cells) [176]. It is possible to nullify the toxic effect of NS5 in the bacteria used for the development of the infectious cDNA clone of ZIKV by sequence modification (i.e., inserting a second copy of the intron sequence after nucleotide position 8882 in NS5) [177], but this procedure does not delete NS5 from virions. For this reason, we performed experiments with a recombinant ZIKV NS5 protein to determine the functional effects of the viral protein on cells.”.
We hope that Reviewer #1 will understand the difficulties of the study and the limitations of clearly associating the analysed cellular events with NS5 by using infectious ZIKV-NS5+ virions.
We would like to thank Reviewer #1 for all these comments, which have improved our work.

Reviewer 2 Report
Comments and Suggestions for Authors
This manuscript uses a transfection approach to analyze the interplay between ZIKV NS5 protein and the cellular HDAC 6 factor in the context of autophagy, tubulin acetylation and IFN responses. Overall the data confirm and extend previous observations with NS5 and I believe that the study will of interest to the field. I do have several suggestions for polishing the manuscript and providing more convincing evidence for several conclusions that are drawn:
Major Points:
1. Fig. 2c: the authors conclude that NS5 promotes tubulin acetylation in a dose dependent fashion – however the 4 of the 5 doses used (1, 1.5. 2 and 3) all report ~ the same relative acetylation. I would recommend repeating this experiment with more values in the linear range of the effect to provide better support for the stated conclusion.
2. Fig. 3d: Overexpressed proteins can sometimes do non-physiological things in cells. As a control, does HDAC6 influence the expression of an unrelated myc-tagged protein (e.g. formally demonstrate that NS5 is a specific target of the cellular factor).
3. Fig. 4a: More convincing data are needed to support the conclusion that the ZIKV NS5 protein negatively alters p62-autophagy flux in cells coexpressing the HA-HDAC6-deltaBUZ construct (Figure 4 a, comparing lanes 2 and 4 with control lane 1). Lane 4 needs to be compared to lane 3 in my opinion (and the change in p62 levels is not convincingly different than the control in the data that are shown).
Minor Points:
1. Methods - Line 201: add ‘poly’ to the phrase: 0.45 um-vinylidene difluoride (PVDF).
2. Fig 1 legend: Please state the number of independent experimental replicates that were used to generate the data/stats in the relevant figure panels.
3. Lines 339-340: while I appreciate the implications, it is not appropriate to conclude anything about NS5 functions in a viral life cycle from a transfection experiment in the absence of viral infection.
4. Line 422: the text states ‘ as measured by the stability of the p62 protein’ – but accumulation of p62 is being measured in the assay, not protein stability per se.
5. Fig. 3a: the x axis of the graph refers to lanes of an individual gel on the left – but the legend states that the data were generated from 4 independent experiments. Thus I would recommend that the authors not refer to gel lanes per se on the graph to better represent how the histogram was generated.
6. Line 665: please clarify: … particularly impairing genomic RNA sensing at its 5′ untranslated region 665 (UTR) capped by RIG-I…. RIG-I, of course, is not an RNA capping enzyme.
7. Line 675: degradative is misspelled (degratative)
8. Fig. 5A: while the response is indeed dramatic, the data do not show that the IFN response is ‘completely abrogated’ as concluded by the authors.
Comments on the Quality of English Language
see above
Author Response
Reply to Reviewer #2:
Comments and Suggestions for Authors
This manuscript uses a transfection approach to analyze the interplay between ZIKV NS5 protein and the cellular HDAC 6 factor in the context of autophagy, tubulin acetylation and IFN responses. Overall the data confirm and extend previous observations with NS5 and I believe that the study will of interest to the field.
We thank Reviewer #2 for this summary and for the positive comments on our work.
I do have several suggestions for polishing the manuscript and providing more convincing evidence for several conclusions that are drawn:
Major Points:
- 2c: the authors conclude that NS5 promotes tubulin acetylation in a dose dependent fashion – however the 4 of the 5 doses used (1, 1.5. 2 and 3) all report ~ the same relative acetylation. I would recommend repeating this experiment with more values in the linear range of the effect to provide better support for the stated conclusion.
We thank Reviewer #2 for this remark.
When we quantified the effect of NS5 on MT acetylation, in some experiments, we observed quite similar effects using 1.5 mg and 2 mg of cDNA from the NS5 construct, but there was a dose-response effect when comparing 0.5, 1/1.5, 2 and 3 mg of cDNA from the NS5 construct in other experiments. The amount of cDNA used in the dose-response experiments was within this range since the addition of more than 3 mg of cDNA from the NS5 construct is toxic to cells.
Figure 2c shows the Ac-tub/a-tub ratio determined by western blot analysis, and we have added three new experiments showing the dose-response effect of NS5 on the acetylation of MTs (new Western blot biochemical analysis). The quantification of these new data has been included in the histogram of Figure 2c, and new Western blot replicates 7, 8 and 9 are presented in supplementary Figure S2C.
- 3d: Overexpressed proteins can sometimes do non- physiological things in cells. As a control, does HDAC6 influence the expression of an unrelated myc-tagged protein (e.g. formally demonstrate that NS5 is a specific target of the cellular factor).
We thank Reviewer #2 for this remark.
The results of the present study suggest that the HDAC6-associated autophagy activity targets either myc- (Figures 2, 3 and 4) or HA-tagged (Figure 5) NS5. Likewise, HA- or DsRed-tagged HDAC6 has a degradative effect on NS5-myc, which is lost when HDAC6 lacks key domains associated with autophagy activity (i.e., dm (hdacs) and DBUZ mutants of the enzyme) or when this activity is inhibited by 3-MA. The results obtained in cells in which endogenous HDAC6 was knocked down by a specific siRNA indicate that the HA or DsRed tag in HDAC6 does not cause HDAC6 to target the ZIKV NS5 protein. Therefore, we believe that the degradative effect exerted by the HA-wt-HDAC6 construct or endogenous HDAC6 on NS5 is independent of the tag present on NS5 and HDAC6.
According to the remark raised by Reviewer #2 about showing data with an unrelated myc-tagged protein, we have experienced HIV-1 viral proteins and cell restriction factors, such as Vif and A3G, respectively (Valera MS, et al. The HDAC6/APOBEC3G complex regulates HIV-1 infectiveness by inducing Vif autophagic degradation. Retrovirology. 2015 Jun 24;12:53. doi: 10.1186/s12977-015-0181-5). In this work, we demonstrated that HDAC6 forms a complex with A3G (either myc- or Flag-tagged), thereby protecting the A3G protein from its HIV-1 Vif-mediated proteasome clearance. In contrast, HDAC6 targets HIV-1 Vif tagged with HA. In this work, HDAC6 protected and did not degrade myc-tagged A3G. Therefore, the myc tag is not related to the degradative action of HDAC6 on any myc-tagged protein.
- 4a: More convincing data are needed to support the conclusion that the ZIKV NS5 protein negatively alters p62- autophagy flux in cells coexpressing the HA-HDAC6-deltaBUZ construct (Figure 4 a, comparing lanes 2 and 4 with control lane 1). Lane 4 needs to be compared to lane 3 in my opinion (and the change in p62 levels is not convincingly different than the control in the data that are shown).
We would like to thank Reviewer #2 for this remark.
In Figure 4a and in the histogram presented in the right part of the figure, we analysed the effect exerted by a mutant of HDAC6 lacking the C-terminal BUZ domain (HDAC6-DBUZ) on NS5. Lane 3 corresponds to an experimental condition in which the cells did not overexpress NS5. Therefore, we only compared lanes 2 and 4 in the histogram of Figure 4a.
In terms of p62, as noted by Reviewer #2, the results presented in Figures 2d and 2e and Figure 4a (lane 2) show the clear effect of the ZIKV NS5 protein on enhancing the expression of the p62 autophagic protein. In the case of Figure 4a, data presented in lane 3 appears to show more p62 than in lane 4, but this is due to the amount of total a-tubulin that has been charged in this lane of the electrophoretic gel (SDS‒PAGE) of proteins. The data needed to be compared considering the total amount of protein charged by lane of the electrophoretic gel (SDS‒PAGE). The HA-HDAC6-DBUZ construct presents a defect in promoting p62-associated autophagy due to the lack of its BUZ domain, as we have discussed in the manuscript (e.g., lines 626-636 of the former Ms) and previously reported (Refs. [100,101,106] of the former and new Ms). The overexpression of this HDAC6 mutant competes with the activity of the endogenous enzyme, resulting in an increase in the expression of the p62 protein (lanes 3 and 4) compared with that in the control conditions (lane 1). Therefore, experimental conditions in cells overexpressing NS5, HA-HDAC6-DBUZ or both constructs are aligned with the impairment of the p62-autophagic flux, which should correspond to higher levels of expression of the p62 protein (lanes 2-4) compared with control cells (lane 1). Hence, and following the question raised by Reviewer #2 about p62, we have added a histogram to the right part of Figure 4a, where the data indicate an increase in p62 protein with no significant differences in p62/a-tubulin values (data are represented by the total protein charged on each lane) between cells expressing NS5, HA-HDAC6-DBUZ or both constructs (lanes 2-4), but this increase was significantly different from that of the control cells (lane 1).
We have therefore commented on these aspects in the Results section (lines 639 - 646), as follows:
“In fact, the expression of the ZIKV NS5 protein, the HDAC6-DBUZ mutant or both constructs increased the expression of the p62 protein (Figure 4a, lanes 2-4, quantified in the associated p62 histogram) compared to the level detected in control cells (Figure 4a, control lane 1, quantified in the associated p62 histogram). Therefore, experimental conditions in cells overexpressing NS5, HA-HDAC6-DBUZ or both constructs are aligned with the impairment of the p62-autophagy flux, which should correspond to higher levels of ex-pression of the p62 protein (Figure 4a, lanes 2-4) compared with control cells (Figure 4a, lane 1).”.
Minor Points:
- Methods - Line 201: add ‘poly’ to the phrase: 0.45 um- vinylidene difluoride (PVDF).
We would like to thank Reviewer #2 for this correction, which we have made as suggested and indicated by the red text.
- Fig 1 legend: Please state the number of independent experimental replicates that were used to generate the data/stats in the relevant figure panels.
We would like to thank Reviewer #2 for this remark.
In Figures 1a and 1b, we have indicated in the former version of the manuscript that “A representative of two experiments is shown (see associated data in Figure S1A or S1B, respectively)”.
In Figures 1e and 1f, we have now indicated the n value of these analyses, thus indicating “per 80 cells counted” (see red text).
- Lines 339-340: while I appreciate the implications, it is not appropriate to conclude anything about NS5 functions in a viral life cycle from a transfection experiment in the absence of viral infection.
We would like to thank Reviewer #2 for this remark.
Therefore, in the revised version of the manuscript (line 340), we have removed this part of the final paragraph “is an integral part of the viral life cycle.”, to avoid any misinterpretation of the data presented.
- Line 422: the text states ‘ as measured by the stability of the p62 protein’ – but accumulation of p62 is being measured in the assay, not protein stability per se.
We would like to thank Reviewer #2 for this remark.
In the revised version of the manuscript (line 422), we have modified this part of the text as follows:
“… as measured by the level of expression of the p62 protein, a marker…”.
- 3a: the x axis of the graph refers to lanes of an individual gel on the left – but the legend states that the data were generated from 4 independent experiments. Thus I would recommend that the authors not refer to gel lanes per se on the graph to better represent how the histogram was generated.
We would like to thank Reviewer #2 for this remark.
We have removed lanes 2 and 4 from the x axis legend to the histogram of Figure 3a.
- Line 665: please clarify: ... particularly impairing genomic RNA sensing at its 5′ untranslated region 665 (UTR) capped by RIG- I.... RIG-I, of course, is not an RNA capping enzyme.
We would like to thank Reviewer #2 for this remark.
In the revised version of the manuscript (new line 676), we have modified the term “capped” to “recognized”.
Although we have not intended to present RIG-I as an RNA capping enzyme, we understand the potential misunderstanding of the use of this term.
- Line 675: degradative is misspelled (degratative)
We would like to thank Reviewer #2 for this remark and correction.
We have now indicated “degradative” (line 685).
- 5A: while the response is indeed dramatic, the data do not show that the IFN response is ‘completely abrogated’ as concluded by the authors.
We would like to thank Reviewer #2 for this remark.
We have changed “completely abrogated” to “strongly diminished” (line 699).
We would like to thank Reviewer #2 for all these comments, which have improved our work.

Reviewer 3 Report
Comments and Suggestions for Authors
Several ZIKV proteins have been extensively studied regarding their roles in viral pathology, biological functions, and evasion of the immune system. In this study, the authors showed that the ZIKV NS5 protein influences the dynamics of the tubulin cytoskeleton and the flux of p62-associated autophagy, both of which are processes regulated by the antiviral factor HDAC6. While the findings of this study are intriguing, there are certain concerns that warrant further investigation, as outlined below.
1. All experiments utilized ectopic expression of ZIKV NS5. To investigate HDAC's role as an anti-NS5/ZIKV factor, the authors should conduct some experiments using Zika virus in combination with wild-type and mutant HDAC6.
2. In Figure 2c, please quantify the levels of Ac-tubulin and calculate the relative ratio of Ac-tubulin/alpha-tubulin.
3. In Figure 3a, there are no HDAC6 signals evident in lane 1 and lane 2. Is this absence due to issues with the antibody or the lack of endogenous expression of HDAC6?
4. Regarding Figure 4a, since HA-HDAC6-DBUZ is expected to enhance the stabilization of the autophagy marker p62, the levels of p62 should theoretically be higher in lane 3 and lane 4 compared to lane 1 and lane 2. However, the p62 protein level appears lower in lane 4 compared to lane 2. Please provide insights on this unexpected result.
5. On Line 435, there is a typo. Please rectify "wyld-type" to "wild-type."
6. Some citation formats in the references are incorrect. For instance, journal information is missing in Ref 181 and 182. Please ensure all references are provided with the correct format.
Comments on the Quality of English LanguageThe manuscript exhibits a smooth and coherent use of the English language.
Author Response
Reply to Reviewer #3:
Comments and Suggestions for Authors:
Several ZIKV proteins have been extensively studied regarding their roles in viral pathology, biological functions, and evasion of the immune system. In this study, the authors showed that the ZIKV NS5 protein influences the dynamics of the tubulin cytoskeleton and the flux of p62-associated autophagy, both of which are processes regulated by the antiviral factor HDAC6.
We thank Reviewer #3 for this summary and for the positive comments on our work.
While the findings of this study are intriguing, there are certain concerns that warrant further investigation, as outlined below.
- All experiments utilized ectopic expression of ZIKV NS5. To investigate HDAC's role as an anti-NS5/ZIKV factor, the authors should conduct some experiments using Zika virus in combination with wild-type and mutant HDAC6.
We thank Reviewer #3 for this remark.
As we have stated in the manuscript (former lines 736-740), the caged reorganization of MTs has been reported in ZIKV-infected cells (Ref. [88] by M. Cortese, et al. Cell Rep 2017 Vol. 18 Issue 9 Pages 2113-2123. DOI: 10.1016/j.celrep.2017.02.014), which surround the viral replication factory close to the endoplasmic reticulum (ER). In this work, the use of the virus avoids establishing a clear-cutting association of the rearrangements observed on MTs with a particular ZIKV protein. Likewise, the authors did not associate the effects observed on MTs with any viral factor or, for example, the NS5 protein. A virion lacking NS5 is required to determine NS5 function together with the chance that no other ZIKV protein or viral factor contributes to the observed effects. Due to the importance of the ZIKV NS5 protein for viral replication (Elshahawi H, et al. Importance of Zika Virus NS5 Protein for Viral Replication. Pathogens. 2019 Sep 30;8(4):169. doi: 10.3390/pathogens8040169; Zhao B, et al. Structure and function of the Zika virus full-length NS5 protein. Nat. Commun. 2017;8:14762. doi: 10.1038/ncomms14762), it would be difficult to obtain delta-NS5 virions, and deleterious point mutations in NS5 could abrogate ZIKV replication or enhance its toxicity (i.e., NS5 2634 and 3328 mutations), making it very difficult to obtain viral progeny (i.e., this virus could be toxic to bacteria and packaging cells) (Collette NM, et al. Single Amino Acid Mutations Affect Zika Virus Replication In Vitro and Virulence In Vivo. Viruses. 2020 Nov 12;12(11):1295. doi: 10.3390/v12111295). It is possible to nullify the toxic effect of NS5 by sequence modification (i.e., inserting a second copy of the intron sequence after nt position 8882 in NS5) in the bacteria used for the development of the infectious cDNA clone of ZIKV (Tsetsarkin KA, et al. A Full-Length Infectious cDNA Clone of Zika Virus from the 2015 Epidemic in Brazil as a Genetic Platform for Studies of Virus‒Host Interactions and Vaccine Development. mBio. 2016;7:e01114-16. doi: 10.1128/mBio.01114-16), but this procedure does not delete NS5 from virions.
For these reasons, we performed experiments with a recombinant ZIKV NS5 protein to determine the functional effects of the viral protein on cells. We believe that this is the first time that the caged effect observed on MTs and the increase in the level of expression of the p62 protein are proposed to be exerted by the ZIKV NS5 protein, as well as the characterization of the effect exerted by HDAC6-associated autophagy on NS5 degradation and IFN production.
In this sense, we have added text to the Discussion section (new lines 756-770), as follows:
“It is important to consider that the use of an infectious ZIKV avoids establishing a clear-cutting association of the rearrangements observed on MTs with a particular ZIKV protein, as previously reported [88]. The authors did not associate the effects observed on the cytoskeleton with any viral factor, such as the NS5 protein. A virion lacking NS5 is required to determine NS5 function together with the chance that no other ZIKV protein or viral factor contributes to the observed effects. Due to the importance of the ZIKV NS5 protein for viral replication [66,170-172], it would be difficult to produce and expand delta-NS5 virions, and deleterious mutations in NS5 could abrogate ZIKV replication [173-175] or enhance its toxicity (i.e., NS5 2634 and 3328 mutations) [176], as it is not easy to obtain viral progeny (i.e., this virus could be toxic to bacteria and packaging cells) [176]. It is possible to nullify the toxic effect of NS5 in the bacteria used for the development of the infectious cDNA clone of ZIKV by sequence modification (i.e., inserting a second copy of the intron sequence after nucleotide position 8882 in NS5) [177], but this procedure does not delete NS5 from virions. For this reason, we performed experiments with a recombinant ZIKV NS5 protein to determine the functional effects of the viral protein on cells.”.
We hope that Reviewer #3 will understand the difficulties of the study and the limitations of clearly associating the analysed cellular events with NS5 by using infectious ZIKV-NS5+ virions.
- In Figure 2c, please quantify the levels of Ac-tubulin and calculate the relative ratio of Ac-tubulin/alpha-tubulin.
We would like to thank Reviewer #3 for this remark.
As suggested by Reviewer #3, the Ac-tub/a-tub ratio determined by western blot analysis is presented in the new Figure 2c.
Moreover, as suggested by Reviewer #2, we have added three new experiments on the dose-response effect of NS5 on the acetylation of MTs (new Western blot biochemical analysis). The quantification of these new data has been included in the histogram of Figure 2c, and new Western blot replicates 7, 8 and 9 are presented in supplementary Figure S2C.
- In Figure 3a, there are no HDAC6 signals evident in lane 1 and lane 2. Is this absence due to issues with the antibody or the lack of endogenous expression of HDAC6?
We would like to thank Reviewer #3 for this remark.
We apologize for this misunderstanding, since we have not indicated that the HDAC6 bands of this western blot were revealed by using an anti-HA antibody that only labels HA-tagged HDAC6 constructs. Lanes 1 and 2 correspond to cells that did not overexpress HA-tagged HDAC6; therefore, no band was detected by the anti-HA antibody. We have now included “HDAC6 construct (a-HA)” in this western blot analysis, as indicated in Figure 3d.
- Regarding Figure 4a, since HA-HDAC6-DBUZ is expected to enhance the stabilization of the autophagy marker p62, the levels of p62 should theoretically be higher in lane 3 and lane 4 compared to lane 1 and lane 2. However, the p62 protein level appears lower in lane 4 compared to lane 2. Please provide insights on this unexpected result.
We would like to thank Reviewer #3 for this remark.
In Figure 4a and in the histogram presented in the right part of the figure, we analysed the effect exerted by a mutant of HDAC6 lacking the C-terminal BUZ domain (HDAC6-DBUZ) on NS5. Lane 3 corresponds to an experimental condition in which the cells did not overexpress NS5. Therefore, we only compared lanes 2 and 4 in the histogram of Figure 4a.
In terms of p62, as noted by Reviewer #2, the results presented in Figures 2d and 2e and Figure 4a (lane 2) show the clear effect of the ZIKV NS5 protein on enhancing the expression of the p62 autophagic protein. In the case of Figure 4a, data presented in lane 3 appears to show more p62 than in lane 4, but this is due to the amount of total a-tubulin that has been charged in this lane of the electrophoretic gel (SDS‒PAGE) of proteins. The data needed to be compared considering the total amount of protein charged by lane of the electrophoretic gel (SDS‒PAGE). The HA-HDAC6-DBUZ construct presents a defect in promoting p62-associated autophagy due to the lack of its BUZ domain, as we have discussed in the manuscript (e.g., lines 626-636 of the former ms) and previously reported (Refs. [100,101,106] of the former and new ms). The overexpression of this HDAC6 mutant competes with the activity of the endogenous enzyme, resulting in an increase in the expression of the p62 protein (lanes 3 and 4) compared with that in the control conditions (lane 1). Therefore, experimental conditions in cells overexpressing NS5, HA-HDAC6-DBUZ or both constructs are aligned with the impairment of the p62-autophagic flux, which should correspond to higher levels of expression of the p62 protein (lanes 2-4) compared with control cells (lane 1). Hence, and following the question raised by Reviewer #3 about p62, we have added a histogram to the right part of Figure 4a, where the data indicate an increase in p62 protein with no significant differences in p62/a-tubulin values (data are represented by the total protein charged on each lane) between cells expressing NS5, HA-HDAC6-DBUZ or both constructs (lanes 2-4), but this increase was significantly different from that of the control cells (lane 1).
We have therefore commented on these aspects in the Results section (lines 639 - 646), as follows:
“In fact, the expression of the ZIKV NS5 protein, the HDAC6-DBUZ mutant or both constructs increased the expression of the p62 protein (Figure 4a, lanes 2-4, quantified in the associated p62 histogram) compared to the level detected in control cells (Figure 4a, control lane 1, quantified in the associated p62 histogram). Therefore, experimental conditions in cells overexpressing NS5, HA-HDAC6-DBUZ or both constructs are aligned with the impairment of the p62-autophagy flux, which should correspond to higher levels of ex-pression of the p62 protein (Figure 4a, lanes 2-4) compared with control cells (Figure 4a, lane 1).”.
- On Line 435, there is a typo. Please rectify "wyld-type" to "wild-type."
We would like to thank Reviewer #3 for this remark and correction.
We have now stated “wild-type”.
- Some citation formats in the references are incorrect. For instance, journal information is missing in Ref 181 and 182. Please ensure all references are provided with the correct format.
We would like to thank Reviewer #3 for this remark and correction.
We have corrected the information associated with these two references (former references 181 and 182, now references 189 and 190, respectively) and checked the format of the references in the manuscript.
We would like to thank Reviewer #3 for all these comments, which have improved our work.

Round 2
Reviewer 1 Report
Comments and Suggestions for Authors
The manuscript is now suitable for publication
Reviewer 2 Report
Comments and Suggestions for Authors
The authors have provided an in-depth response to all of the points previously raised. I find the revised manuscript to be improved and convincing.
Reviewer 3 Report
Comments and Suggestions for Authors
The revised MS has relieved my concern and is acceptable now.